

# Multicomponent one-dimensional quantum droplets across the mean-field stability regime

I. A. Englezos[1*], P. Schmelcher[1,2] and S. I. Mistakidis[3]

**1** Center for Optical Quantum Technologies, Department of Physics, University of Hamburg, Luruper Chaussee 149, 22761 Hamburg Germany
**2** The Hamburg Centre for Ultrafast Imaging, University of Hamburg, Luruper Chaussee 149, 22761 Hamburg, Germany
**3** Department of Physics and LAMOR, Missouri University of Science and Technology, Rolla, MO 65409, USA

* ilias.englezos@uni-hamburg.de

## Abstract

The Lee-Huang-Yang (LHY) energy correction at the edge of the mean-field stability regime is known to give rise to beyond mean-field structures in a wide variety of systems. In this work, we analytically derive the LHY energy for two-, three- and four-component one-dimensional bosonic short-range interacting mixtures across the mean-field stability regime. For varying intercomponent attraction in the two-component setting, quantitative deviations from the original LHY treatment emerge being imprinted in the droplet saturation density and width. On the other hand, for repulsive interactions an unseen early onset of phase-separation occurs for both homonuclear and heteronuclear mixtures. Closed LHY expressions for the fully-symmetric three- and four-component mixtures, as well as for mixtures comprised of two identical components coupled to a third independent component are provided and found to host a plethora of mixed droplet states. Our results are expected to inspire future investigations in multicomponent systems for unveiling exotic self-bound states of matter and unravel their nonequilibrium quantum dynamics.



# 1  Introduction

Since the first experimental realization of the Bose-Einstein condensate [1–3], outstanding progress has been sealed in terms of realizing, monitoring and controlling multicomponent atomic gases [4–6]. This is largely owed to exploiting Feshbach resonances [7] and trapping techniques [8, 9] to manipulate the interatomic interaction strength or range and the external confinement landscape respectively. Starting from single-component bosonic gases, a multitude of correlated phases beyond the weakly interacting mean-field (MF) limit were observed, such as the (Super-) Tonks-Girardeau states for strong (attractions [10, 11]) repulsions [12–15]. Turning to bosonic mixtures arguably enriched correlated phases, already for weak interactions, occur due to the competition between the intra- and inter-species interactions. This has lead, for instance, to phase separation [16–18], when the interspecies repulsion surpasses the average intraspecies one, bubble phases at the immiscibility threshold [19], or the recently realized quantum droplets [20–23] when the interspecies attraction slightly overcomes the intraspecies repulsions.

The emergence of the self-bound liquid-type configurations known as quantum droplets in contact interacting three-dimensional (3D) bosonic mixtures [24, 25], as well as in dipolar gases [26, 27], is a manifestation of the impact of quantum fluctuations. These may be captured by the Lee-Huang-Yang (LHY) energy [20, 28], representing the first-order correction to the Bogoliubov MF theory, which arrests the expected collapse of the ensuing attractive system at the edge of the MF stability regime [5]. Several properties of these intriguing many-body bound states of matter have already been explored. These refer, exemplarily, to their spectrum [29, 30], collective excitations [20, 29, 31], collisions [31, 32] and coexistence with nonlinear soliton [33, 34] and vortex [35–38] states, within the context of the LHY theory [21, 22, 39], see also Refs. [40–43] for the exposition of beyond-LHY effects by deploying *ab-initio* methods. Experimentally, quantum droplets have been observed in 3D in both homonuclear [24, 25, 44] and heteronuclear [45–47] mixtures. One of the main challenges faced by these experimental efforts stems from the relatively short droplet lifetimes caused (primarily) by three-body recombination [48]. These lifetimes are increased in relevant 1D settings [49, 50] due to lower densities but also in heteronuclear mixtures [45].

A particularly interesting property is the dependence of the LHY term on the dimensionality [50–52], in contrast to MF interactions [6]. In 1D that we focus herein, the LHY term is attractive [53] instead of being repulsive as in 3D, significantly changing the droplet parametric region and its phenomenology. Concretely, the 1D geometry causes the droplet parametric

regions to i) not feature collapse, ii) host stable bright-soliton solutions [54–56] and iii) coincide with the respective MF stability regimes. Hence, 1D platforms share the premise of hosting, comparatively easier to control, enriched droplet phases even more in the genuine two- [57–61] and three-component [62] settings. Additionally, quantum droplets and more generally the impact of quantum fluctuations in 1D can be studied throughout the MF stability regime. However, the majority of the current investigations have been motivated by the correspondence to the 3D case, and thus solely focused on the attractive threshold of the MF stability regime. Hence, a general framework being able to describe droplets across the MF stability region but also heteronuclear mixtures in 1D is still lacking. Moreover, extraction and investigation of the LHY energy in higher-component setups is an open question. Indeed, previous studies on three-component mixtures in 1D have been focused on the case of a single-impurity in the third component [63–65] or a MF density coupling between the droplet and the impurities [66,67]. Instead, appropriate inclusion of the LHY term among all components may be able to unveil unseen self-bound structures.

To address the above-discussed open problems, we derive the LHY energy (in a closed form whenever possible) and associated extended Gross-Pitaevskii equations (eGPEs) for two-, three- and four-component 1D bosonic mixtures based on the perturbative Bogoliubov treatment. Previous treatments restricted to two-component settings [53], can be obtained as a limiting case of our framework and are restricted to the 3D wave collapse threshold. In contrast to those, which (for convenience) we dub herein as "original", our exact approach remains valid throughout the 1D MF stability regime. Namely, it encompasses both attractive and repulsive interspecies interactions and interestingly it is able to tackle homonuclear as well as heteronuclear two-component settings. Direct comparisons of our approach to the approximate one in the attractive intercomponent interaction regime reveals quantitative deviations. These are imprinted as a reduced droplet saturation density and increased width in the exact case. Turning to the repulsive side, our results predict an early onset of phase-separation for both homonuclear and heteronuclear mixtures which is absent within the approximate framework.

Next, we provide exact closed form expressions for the LHY energies and ensuing eGPEs of a three-component mixture with either three or two identical components. It is explicated that such settings offer a broad range of possibilities for creating mixed droplet phases that are absent in lower-component setups. Additionally, a closed form LHY expression is extracted for the fully symmetric four-component mixture, constituting the higher-component mixture for which the Bogoliubov modes may be derived analytically in the general case. We present the derivations in a detailed and structured manner aiming to provide a useful resource for researchers entering the rapidly expanding field of quantum droplets and in general attractively interacting bosonic setups.

This work is structured as follows. In Section 2, we introduce the general multicomponent bosonic mixture featuring short-range contact interactions, the Bogoliubov transformation, and the original approximate two-component LHY theory being valid at the edge of the MF stability regime. Section 3 is devoted to the derivation and solution of the exact eGPEs for the genuine two-component mixture across the entire MF stability regime. This treatment is generalized to the three-component (Sec. 4) and the four-component (Sec. 5) bosonic mixture. Concluding remarks and perspectives for future investigations are offered in Sec. 6. In Appendix A, we provide further details regarding the three-component mixture stability conditions. Appendix B discusses the existence of asymmetric (domain-wall like) droplet distributions in the phase-separated regime.

## 2 Basic settings and background

### 2.1 Multicomponent bosonic droplet setting

We consider a multicomponent bosonic mixture consisting of $N_\sigma$ atoms of mass $m_\sigma$ (where $\sigma = A, B, C, D, \ldots$) residing in 1D free space and featuring periodic boundary conditions. Notice, however, that our results (to be presented below) persist also in the case of a box potential provided that its length is sufficiently large and does not impact the edges of the discussed self-bound droplet configurations. The mixture is assumed to be close to zero temperature, where $s$-wave scattering dominates [68]. Hence, interparticle interactions are represented by contact potentials. Their effective coupling strengths are intracomponent repulsive (i.e. $g_{\sigma\sigma} \equiv g_\sigma > 0$) and intercomponent attractive ($g_{\sigma\sigma'} < 0$) or repulsive ($g_{\sigma\sigma'} > 0$). In a corresponding experiment, they can be adjusted either via Fano-Feshbach resonances [69, 70] through the 3D scattering length or through confinement induced resonances [68] by modifying the transverse trap frequency, $\omega_\perp$. This is significantly tight such that transverse excitations are prohibited and the atomic motion is restricted across the elongated $x$-direction as in corresponding 1D experiments [71].

The many-body Hamiltonian in second quantized form reads

$$H = \int dx \left( \sum_\sigma \left( \Psi_\sigma^\dagger \hat{h}_\sigma \Psi_\sigma + \frac{g_\sigma}{2} \Psi_\sigma^\dagger \Psi_\sigma^\dagger \Psi_\sigma \Psi_\sigma + \sum_{\sigma' > \sigma} \frac{g_{\sigma\sigma'}}{2} \Psi_\sigma^\dagger \Psi_{\sigma'}^\dagger \Psi_\sigma \Psi_{\sigma'} \right) \right), \tag{1}$$

where $\hat{h}_\sigma = -\frac{\hbar^2}{2m_\sigma} \left( \frac{\partial^2}{\partial x_\sigma^2} \right)$ is the underlying single-particle hamiltonian and $\Psi_\sigma(x)$ refers to the field operator annihilating a $\sigma$ species boson at position $x$. In what follows, for computational convenience, we rescale the above Hamiltonian with respect to the energy scale set by $m_A g_A^2 / \hbar^2$. Hence, the length, time, and interaction strengths are given in units of $\hbar^2/(m_A g_A)$, $\hbar^3/(m_A g_A^2)$, and $g_A$ respectively.

### 2.2 Bogoliubov transformation

For an arbitrary set of $n$ bosonic operators $A^\dagger = (\hat{\alpha}_1^\dagger, \hat{\alpha}_2^\dagger, \ldots, \hat{\alpha}_n^\dagger, \hat{\alpha}_1, \hat{\alpha}_2, \ldots, \hat{\alpha}_n)$ the general Bogoliubov transformation is defined by $A = UB$. Here, $B^\dagger = (\hat{\beta}_1^\dagger, \hat{\beta}_2^\dagger, \ldots, \hat{\beta}_n^\dagger, \hat{\beta}_1, \hat{\beta}_2, \ldots, \hat{\beta}_n)$ are the Bogoliubov operators and $U = \begin{pmatrix} M & N \\ N^* & M^* \end{pmatrix}$ is the Bogoliubov transformation matrix, where $M, N$ are $n \times n$ matrices [72]. Demanding that both $A$ and $B$ satisfy bosonic commutation relations translates to $\mathcal{M}_b = [A, A^\dagger] = [B, B^\dagger] = \begin{pmatrix} I_n & 0 \\ 0 & -I_n \end{pmatrix}$, with $I_n$ denoting the $n$-dimensional identity matrix. As such, utilizing the arbitrary Bogoliubov transformation yields

$$\mathcal{M}_b = U \mathcal{M}_b U^\dagger. \tag{2}$$

Thus, demanding that Eq. (2) holds is equivalent to imposing that the Bogoliubov transformation preserves the commutation relations of the set of bosonic operators. However, in this case, it is not required to explicitly derive the constraints on the elements of the transformation matrix $U$, as done in the usual textbook derivation [5, 6].

Another useful property of the Bogoliubov transformation is that it can be used to diagonalize a bilinear (in terms of the arbitrary set of bosonic operators $A$) operator, such as $\hat{H} = A^\dagger H A$. Here, $\hat{H}$ is an one-body operator (expressed in second quantized form) with respect to $\hat{a}_i$, $\hat{a}_i^\dagger$ and $H$ is its corresponding matrix representation. Accordingly, it holds that $\hat{H} = A^\dagger H A = B^\dagger U^\dagger H U B = B^\dagger \mathcal{M}_b U^{-1} \mathcal{M}_b H U B$. Then, by requiring that the Bogoliubov trans-

formation[1] diagonalizes $\hat{H}$ in terms of the bilinear form with respect to $B$ or equivalently diagonalizes the matrix $\mathcal{H}_M = \mathcal{M}_b H$, as $U^{-1}\mathcal{H}_M U = D$, where $D$ is a diagonal matrix, we obtain the eigenvalues of $\mathcal{H}_M$ through the characteristic equation

$$\det(\mathcal{M}_b H - \lambda I_{2n}) = 0. \tag{3}$$

This leads to $\mathcal{M}_b D = \mathrm{diag}(\lambda_1, \ldots, \lambda_n, -\lambda_{n+1}, \ldots, -\lambda_{2n})$ and therefore the operator $\hat{H}$ takes the form

$$\hat{H} = B^\dagger \mathcal{M}_b D B = \sum_{i=1}^{n} \left( \lambda_i \hat{\beta}_i^\dagger \hat{\beta}_i - \lambda_{n+i} \hat{\beta}_i \hat{\beta}_i^\dagger \right) = \sum_{i=1}^{n} (\lambda_i - \lambda_{n+i}) \hat{\beta}_i^\dagger \hat{\beta}_i - \sum_{i=1}^{n} \lambda_{n+i}. \tag{4}$$

In the last step, we have used that $[\beta_i, \beta_i^\dagger] = 1$. In this sense, $h_0 = -\sum_{i=1}^{n} \lambda_{n+i}$ is the eigenvalue of $\hat{H}$ corresponding to the quasi-particle vacuum in terms of the bosonic quasi-particle operators $(\beta_i, \beta_i^\dagger)$, while $h_i = \lambda_i - \lambda_{n+i}$ are the eigenvalues corresponding to the quasi-particle excitations.

## 2.3 Original two-component LHY treatment

For a two-component, homonuclear ($m_A = m_B \equiv m$), 1D mixture the MF stability regime is given by the condition $g_{AB}^2 \leq g_A g_B$ [5]. In the attractive limit where $g_{AB} = -\sqrt{g_A g_B}$ holds, taking into account the first-order (LHY) quantum correction it was shown [53] to result in the following coupled set of "approximate" eGPEs describing two-component 1D quantum droplets

$$i\hbar \frac{\partial \Psi_\sigma}{\partial t} = \left( -\frac{\hbar^2}{2m} \frac{\partial^2}{\partial x_\sigma^2} + g_\sigma |\Psi_\sigma|^2 + g_{AB} \left| \Psi_{\sigma' \neq \sigma} \right|^2 - \frac{g_\sigma \sqrt{m}}{\pi \hbar} \sqrt{g_A |\Psi_A|^2 + g_B |\Psi_B|^2} \right) \Psi_\sigma. \tag{5}$$

In these equations, the first terms represent the usual mean-field Gross-Pitaevskii equations (GPEs) and the last term provides the LHY correction. In the symmetric system, where the intracomponent interactions $g_\sigma \equiv g$ and the species densities $n_\sigma \equiv n$, with $N_\sigma = \int n_\sigma dx = \int |\Psi_\sigma|^2 dx$ the components behave equivalently. Accordingly, it has been demonstrated [21,23] that these eGPEs predict a transition of the droplet density from a Gaussian to a flat-top (FT) configuration upon either increasing the particle number ($N$) or decreasing the intercomponent attraction ($|g_{AB}|$) with $0 < -g_{AB} < g$ [53]. The emergent FT structures feature a saturated peak density at [53] $n_0 = \frac{8mg^3}{9\hbar^2\pi^2(g+g_{AB})^2}$, with energy density $\mathcal{E}_0(n = n_0) = -(g + g_{AB})n_0^2$, chemical potential $\mu_0 = \frac{\partial \mathcal{E}_0}{\partial n}(n = n_0) = -(g + g_{AB})n_0$, and healing length $\xi_0 = \sqrt{-\frac{\hbar^2}{2m\mu_0}} = \sqrt{\frac{\hbar^2}{2m(g+g_{AB})n_0}}$.

It is imperative to clarify that the extraction of Eq. (5) is based on the assumption that we are operating at the interaction boundary $g_{AB} = -\sqrt{g_A g_B}$. This is indeed well motivated by the respective 3D setting in which the LHY contribution is repulsive and provides stabilization outside the MF stability regime where droplets emerge [20]. However, in 1D systems the LHY correction turns out to be attractive [23,53], and therefore the droplet configurations are hosted within the MF stability regime. Hence, as we will explicate in detail below, no assumptions (beyond the usual ones deployed for a dilute and weakly interacting gas) are needed, and we can analytically extract eGPEs which are valid throughout the MF stability regime and not only in the boundary as was currently known. This is one of the main results of our work which presents the generalized two-component eGPEs and exposes deviations from the predictions of the "original" ones described by Eq. (5).

---

[1]The Bogoliubov transformation matrix $U$ is not unique. Thus, additional constraints can be imposed by exploiting the remaining (i.e. not the ones required to preserve the bosonic symmetry) degrees-of-freedom.

# 3 Two-component mixture

The Hamiltonian density for a short-range (contact) interacting two-component bosonic mixture in a box of length $L$, can be easily deduced from the general multicomponent Hamiltonian given by Eq. (1) with $\sigma = A, B$ and $\sigma' = A, B \neq \sigma$. A corresponding experimentally relevant system of choice here could be, for instance, two different hyperfine states of $^{39}$K as in the 3D case [24]. Next, we express the field operators in the plane-wave basis as $\hat{\Psi}_A(x) = \frac{1}{\sqrt{L}}\hat{a}_0 + \sum_{k\neq0}\frac{e^{ikx}}{\sqrt{L}}\hat{a}_k$ and $\hat{\Psi}_B(x) = \frac{1}{\sqrt{L}}\hat{b}_0 + \sum_{k\neq0}\frac{e^{ikx}}{\sqrt{L}}\hat{b}_k$, where we have implicitly assumed periodic boundary conditions. Substitution of these operators into the aforementioned two-component Hamiltonian and integration over the box length, leads to the Hamiltonian

$$
\begin{aligned}
\hat{\mathcal{H}} = {}& \frac{g_A}{2L}\hat{a}_0^\dagger\hat{a}_0^\dagger\hat{a}_0\hat{a}_0 + \frac{g_B}{2L}\hat{b}_0^\dagger\hat{b}_0^\dagger\hat{b}_0\hat{b}_0 + \frac{g_{AB}}{L}\hat{a}_0^\dagger\hat{b}_0^\dagger\hat{a}_0\hat{b}_0 + \frac{\hbar^2}{2m_A}\sum_{k\neq0}k^2\hat{a}_k^\dagger\hat{a}_k + \frac{\hbar^2}{2m_B}\sum_{k\neq0}k^2\hat{b}_k^\dagger\hat{b}_k \\
& + \frac{g_A}{2L}\sum_{k\neq0}\big[\hat{a}_0^\dagger\hat{a}_0^\dagger\hat{a}_k\hat{a}_{-k} + \hat{a}_0\hat{a}_0\hat{a}_k^\dagger\hat{a}_{-k}^\dagger + 4\hat{a}_0^\dagger\hat{a}_0\hat{a}_k^\dagger\hat{a}_k\big] \\
& + \frac{g_B}{2L}\sum_{k\neq0}\big[\hat{b}_0^\dagger\hat{b}_0^\dagger\hat{b}_k\hat{b}_{-k} + \hat{b}_0\hat{b}_0\hat{b}_k^\dagger\hat{b}_{-k}^\dagger + 4\hat{b}_0^\dagger\hat{b}_0\hat{b}_k^\dagger\hat{b}_k\big] \\
& + \frac{g_{AB}}{L}\sum_{k\neq0}\big[\hat{a}_0^\dagger\hat{b}_0^\dagger\hat{a}_k\hat{b}_{-k} + \hat{a}_0^\dagger\hat{a}_0\hat{b}_k^\dagger\hat{b}_k + \hat{a}_0^\dagger\hat{b}_0\hat{b}_k^\dagger\hat{a}_k + \hat{b}_0^\dagger\hat{a}_0\hat{a}_k^\dagger\hat{b}_k + \hat{b}_0^\dagger\hat{b}_0\hat{a}_k^\dagger\hat{a}_k + \hat{a}_0\hat{b}_0\hat{a}_k^\dagger\hat{b}_{-k}^\dagger\big] \\
& + \mathcal{O}(\hat{a}_k^3).
\end{aligned}
\tag{6}
$$

Here, terms higher than bilinear in the bosonic operators $\hat{a}_k$, $\hat{b}_k$ are neglected. From this expression, it is possible to readily retrieve the zeroth order (MF) approximation by setting $\hat{a}_0 = \sqrt{N_A}$, $\hat{b}_0 = \sqrt{N_B}$ and neglecting all other terms with $(\hat{a}_{k\neq0}, \hat{b}_{k\neq0})$. Such a process results in the known MF energy density of the two-component setting

$$
\mathcal{E}_{\mathrm{MF}} = \frac{E_{\mathrm{MF}}}{L} = \frac{1}{2}g_A n_A^2 + \frac{1}{2}g_B n_B^2 + g_{AB}n_A n_B.
\tag{7}
$$

To obtain the first-order correction to the MF energy, one needs to account for the depletion of the condensate due to quantum fluctuations. This is achieved by expressing the particle number operators as $N_A = \hat{a}_0^\dagger\hat{a}_0 + \sum_{k\neq0}\hat{a}_k^\dagger\hat{a}_k$ and $N_B = \hat{b}_0^\dagger\hat{b}_0 + \sum_{k\neq0}\hat{b}_k^\dagger\hat{b}_k$. Inserting those into the Hamiltonian of Eq. (6) and defining the set of operators $\Phi^\dagger = (\hat{a}_k^\dagger, \hat{b}_k^\dagger, \hat{a}_{-k}, \hat{b}_{-k})$, we arrive at the bilinear Hamiltonian

$$
\hat{H} = \sum_{k>0}\Phi^\dagger H \Phi + \frac{g_A N_A^2}{2L} + \frac{g_B N_B^2}{2L} + \frac{g_{AB}N_B N_A}{L} - \sum_{k>0}\Big(\frac{\hbar^2 k^2}{2m_A} + \frac{\hbar^2 k^2}{2m_B} + g_A n_A + g_B n_B\Big).
\tag{8}
$$

The operators, $\Phi$, satisfy the bosonic commutation relations, i.e. $\mathcal{M}_b = [\Phi, \Phi^\dagger] = \begin{pmatrix} I_2 & 0 \\ 0 & -I_2 \end{pmatrix}$, where $I_2$ is the $2\times2$ identity matrix. Also, the Hamiltonian matrix

$$
H = \begin{pmatrix} h_A(k) & h_{AB} & g_A n_A & h_{AB} \\ h_{AB} & h_B(k) & h_{AB} & g_B n_B \\ g_A n_A & h_{AB} & h_A(k) & h_{AB} \\ h_{AB} & g_B n_B & h_{AB} & h_B(k) \end{pmatrix},
\tag{9}
$$

where $h_\sigma(k) = \frac{\hbar^2 k^2}{2m_\sigma} + g_\sigma n_\sigma$ and $h_{AB} = g_{AB}\sqrt{n_A n_B}$. Note that the sums in Eq. (8) are now over positive momenta only. As argued in Sec. 2.2, we can diagonalize the Hamiltonian of Eq. (8) in terms of an appropriate Bogoliubov transformation, by solving the

characteristic equation (Eq. (3)) $\det(\mathcal{M}_b H - \omega_k I_4) = 0$ which in this case takes the form $\omega^4 + \beta \omega^2 + \gamma = 0$. The parameters are $\beta = -\epsilon_A^2(k) - \epsilon_B^2(k)$ and $\gamma = \epsilon_A^2(k)\epsilon_B^2(k) - g_{AB}^2 n_A n_B \frac{\hbar^4 k^4}{m_A m_B}$, with $\epsilon_\sigma(k) = \sqrt{\frac{\hbar^4 k^4}{4m_\sigma^2} + \frac{\hbar^2 k^2}{m_\sigma} g_\sigma n_\sigma}$ representing the corresponding single-component Bogoliubov dispersion relations [6]. The solutions of the characteristic polynomial equation yield the underlying dispersion relation for the two-component system in the presence of the first-order quantum correction

$$\omega_\pm^2 = \frac{\epsilon_A^2 + \epsilon_B^2 \pm \sqrt{(\epsilon_A^2 - \epsilon_B^2)^2 + 4g_{AB}^2 n_A n_B \frac{\hbar^4 k^4}{m_A m_B}}}{2}. \tag{10}$$

Finally, utilizing the diagonal form of a bilinear operator within the Bogoliubov transformation prescribed by Eq. (4) and collecting the constant terms in Eq. (8), we find the respective ground state energy

$$E_0 = \frac{g_A N_A^2}{2L} + \frac{g_B N_B^2}{2L} + \frac{g_{AB} N_B N_A}{L} + \sum_{k>0}\left(\omega_+ + \omega_- - \frac{\hbar^2 k^2}{2m_A} - \frac{\hbar^2 k^2}{2m_B} - g_A n_A - g_B n_B\right). \tag{11}$$

Apparently, the first terms in Eq. (11) correspond to the standard MF intra- and inter-component interaction energies respectively, while the sum is the first-order LHY quantum correction for the two-component mixture. To calculate the LHY contribution, we substitute the sum in Eq. (11) with an integral over $k$ according to $\sum_{k>0} \to (L/2\pi)\int_0^{+\infty} dk$. We note in passing that in the 3D case, additional terms ($\propto +\frac{m g_{\sigma\sigma'} n_\sigma}{\hbar^2 k^2}$) enter the corresponding sum over $k$, while the latter has to be substituted with the corresponding 3D integral [6].

## 3.1 Exact closed form solution for the homonuclear mixture

As already discussed above, unlike the 3D case (see Ref. [20]), in 1D it is not necessary to impose the approximation $g_{AB}^2 \approx g_A g_B$ (which leads to Eq. (5)) in the dispersion relation of Eq. (10). In particular, it is possible to calculate exactly the ground state energy for a homonuclear ($m_A = m_B \equiv m$) mixture. Indeed, setting $p := \frac{4(g_{AB}^2 - g_A g_B)n_A n_B}{(g_A n_A + g_B n_B)^2}$, and by employing the change of variables $\hbar k = \sqrt{m(g_A n_A + g_B n_B)}\,y$ to calculate the integral in Eq. (11), the ground state energy density becomes

$$\mathcal{E} = \frac{E_0}{L} = \mathcal{E}_{MF} + \underbrace{\frac{\sqrt{m}(g_A n_A + g_B n_B)^{\frac{3}{2}}}{2\hbar\pi}\mathcal{I}(p)}_{\mathcal{E}_{LHY}}, \tag{12}$$

where the function of the beyond MF energy contribution

$$\mathcal{I}(p) = -\frac{\sqrt{2}}{3}\left((1 - \sqrt{p+1})^{\frac{3}{2}} + (1 + \sqrt{p+1})^{\frac{3}{2}}\right). \tag{13}$$

Importantly, the generalized two-component energy density described by Eq. (12) *is valid for any interaction strength and particle number* for which the MF treatment is valid. Moreover, the LHY energy is real-valued and negative, as long as $-1 \le p \le 0$, i.e. $|g_{AB}| \le \sqrt{g_A g_B}$, so that $p \le 0$ (while $p \ge -1$ is trivially satisfied). Notice that this interaction interval corresponds to the *entire MF stability region*, ranging from the usual droplet threshold ($-g_{AB} \approx \sqrt{g_A g_B}$) all the way to the standard miscibility threshold ($g_{AB} \approx \sqrt{g_A g_B}$) [16,23]. It is also worth mentioning explicitly that, in the limit of $p = 0$, we obtain $\mathcal{I}(p = 0) = -4/3$, i.e. *we recover the original droplet energy* which results in Eq. (5). Moreover, at the edge of the MF stability regime where

$g_{AB}^2 \approx g_A g_B$, it holds that $|p| \ll 1$, leading to $I(p) \approx -\frac{4}{3} - \frac{p}{2}$. Finally, in the opposite limit of an uncoupled mixture ($g_{AB} = 0$) we find $I(p = -1) = -2\sqrt{2}/3$. Hence, in this case the LHY term is reduced by a factor of $\sqrt{2}$. A similar expression was recently derived in the context of a mobile impurity immersed in a droplet host [66].

Having determined the ground state energy of the two-component bosonic mixture in the presence of the LHY correction, we can subsequently derive the exact eGPEs. This is done by evaluating the Euler-Lagrange equations with respect to $n_\sigma$ yielding:

$$i\hbar\frac{\partial \Psi_\sigma}{\partial t} = \left(-\frac{\hbar^2}{2m}\frac{\partial^2}{\partial x^2} + \frac{\partial \mathcal{E}}{\partial n_\sigma}\right)\Psi_\sigma = \left(-\frac{\hbar^2}{2m}\frac{\partial^2}{\partial x^2} + g_\sigma|\Psi_\sigma|^2 + g_{AB}|\Psi_{\sigma'\neq\sigma}|^2 + \frac{\partial \mathcal{E}_{\mathrm{LHY}}}{\partial n_\sigma}\right)\Psi_\sigma. \quad (14)$$

In general, the full eGPEs have a quite complicated form, since

$$\frac{\partial \mathcal{E}_{\mathrm{LHY}}}{\partial n_\sigma} = \frac{3g_\sigma\sqrt{m(g_A n_A + g_B n_B)}}{4\hbar\pi}\mathcal{I}(p) + \frac{\sqrt{m(g_A n_A + g_B n_B)^3}}{2\hbar\pi}\frac{\partial \mathcal{I}}{\partial p}\frac{\partial p}{\partial n_\sigma}, \quad (15)$$

where

$$\frac{\partial \mathcal{I}}{\partial p} = \frac{\sqrt{2}\left(\sqrt{1 - \sqrt{p+1}} - \sqrt{1 + \sqrt{p+1}}\right)}{4\sqrt{p+1}}, \quad (16a)$$

$$\frac{\partial p}{\partial n_\sigma} = \frac{4n_{\sigma'}(g_A g_B - g_{AB}^2)(g_\sigma n_\sigma - g_{\sigma'}n_{\sigma'})}{(g_A n_A + g_B n_B)^3}, \quad (16b)$$

with $\sigma \neq \sigma'$. Notably, the form and subsequent impact of the LHY correction away from the droplet limit ($-g_{AB} \approx \sqrt{g_A g_B}$) and even for repulsive intercomponent interactions (where e.g. bubble configurations have been predicted [19]) is largely unknown. Considering stronger intercomponent interactions outside the MF stability, i.e. $|g_{AB}| > \sqrt{g_A g_B}$ where $p > 0$, results in a complex LHY correction. A common practice in the literature is to simply neglect the imaginary part of $\mathcal{E}$ [20, 62], assuming that it is sufficiently small and hence the growth rate of the resulting instability is slower than e.g. the decay due to three-body recombination. Examining the validity of this treatment by comparing its predictions e.g. with MF computations, *ab-initio* calculations [40–43] or experimental data would be a particularly interesting direction for future studies, especially in 1D. This is because, aside from being more amenable to *ab-initio* calculations, 1D systems do not experience MF collapse in this limit, unlike their 3D counterparts where stable solutions exist only in the presence of the LHY term [20].

An insightful application of our generalized framework is to consider the limiting case of a fully balanced mixture characterized by $g_A = g_B \equiv g$ and $n_A = n_B \equiv n$, and therefore $p = (\frac{g_{AB}}{g})^2 - 1$. Accordingly, the second term in Eq. (15) vanishes and the LHY term becomes $\sim (g^3 n)^{1/2}I(p)$. Thus, in this case, the exact LHY term has the same density dependence as in the "approximate" eGPEs [53] given by Eq. (5), but they are quantitatively different since the exact LHY term depends also on $g_{AB}$. Using the exact energy of the full two-component system given by Eq. (12), we find the minimum of the energy per particle for the aforementioned balanced mixture to occur at the saturation density

$$n_s = \frac{mg^3 I^2(p)}{2\hbar^2\pi^2(g+g_{AB})^2} = \frac{2mg^3\left[1 + 3(g_{AB}/g)^2 + (1-(g_{AB}/g)^2)^{3/2}\right]}{9\hbar^2\pi^2(g+g_{AB})^2}. \quad (17)$$

Here, the energy density $\mathcal{E}(n = n_s) = \frac{-m^2 g^6 I^4(p)}{8\hbar^4\pi^4(g+g_{AB})^3} = -(g + g_{AB})n_s^2$, the chemical potential $\mu_s = \frac{-mg^3 I^2(p)}{2\hbar^2\pi^2(g+g_{AB})} = -(g + g_{AB})n_s$, and the healing length $\xi_s = \sqrt{-\frac{\hbar^2}{2m\mu_s}} = \sqrt{\frac{\hbar^2}{2m(g+g_{AB})n_s}}$.

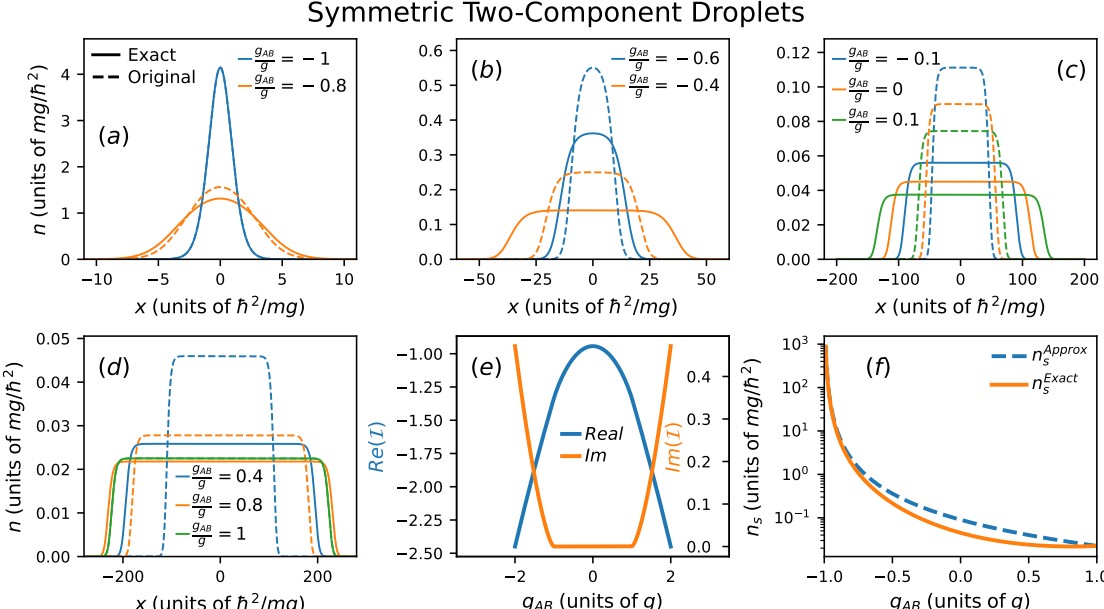

Figure 1: Ground state droplet densities for a symmetric mixture featuring $N_A = N_B = N = 10$, $g_A = g_B = g$, and varying intercomponent interactions $g_{AB}$ (see legends). A plethora of different cases are shown ranging from (a) strong and (b) weak attractions, to (c) weak (nearly suppressed) couplings all the way towards (d) repulsive intercomponent interactions. In all cases, the ground state density obtained within the original given by Eq. (5) (dashed lines) and the exact (solid lines) eGPEs (14) are depicted. Evidently, the original eGPEs consistently predict significantly more localized configurations when $|g_{AB}| \neq g$, while in some cases (e.g. $g_{AB} = -0.6g$) it even fails to adequately capture the droplet's FT density profile. Note that the dashed blue and green lines in panels (a) and (d) respectively are hardly visible since they coincide with the solid ones. (e) Dependence of the real and imaginary parts (see legend) of the LHY energy coefficient ($I(p)$) in Eq. (13) on the interspecies interaction strength. (f) The droplet saturation density in both the exact and original approaches (see legend) with respect to $g_{AB}$.

Apparently, these exact results exhibit quantitative differences from the predictions of the original LHY approach described by Eq. (5). This can be readily deduced, for instance, by comparing them with the approximate saturation density ($n_0$) as well as the chemical potential and healing length discussed in Sec. 2.3 for the balanced mixture. Note that the chemical potentials and energies for all 1D droplet configurations discussed throughout this work, are negative. This supports their self-bound nature in the entire MF stability regime.

To further elucidate the deviations between the original (Eq. (5)) and the exact (Eq. (14)) eGPEs predictions we numerically compute, via imaginary time-propagation, the corresponding ground state droplet densities for the symmetric mixture with $g_A = g_B \equiv g$ and $N_A = N_B \equiv N = 10$. Since, the exact eGPEs (Eq. (14)) are valid throughout the MF stability regime, we present results for the standard droplet regime $g_{AB} < 0$ [Fig. 1(a), (b)], but also for the nearly intercomponent decoupled ($g_{AB} \approx 0$) system [Fig. 1(c)] and even for repulsive interactions within the miscible regime quantified by $0 < g_{AB} \leq g$ [Fig. 1(d)]. The underlying LHY energy, $\mathcal{E}(n = n_s) = -(g + g_{AB})n_s^2$, attains its maximum (or minimum absolute) value for the decoupled system, whilst it decreases when $g_{AB}$ is nonzero. Interestingly, it acquires a relatively small imaginary component for $|g_{AB}| > g$ as shown in Fig. 1(e).

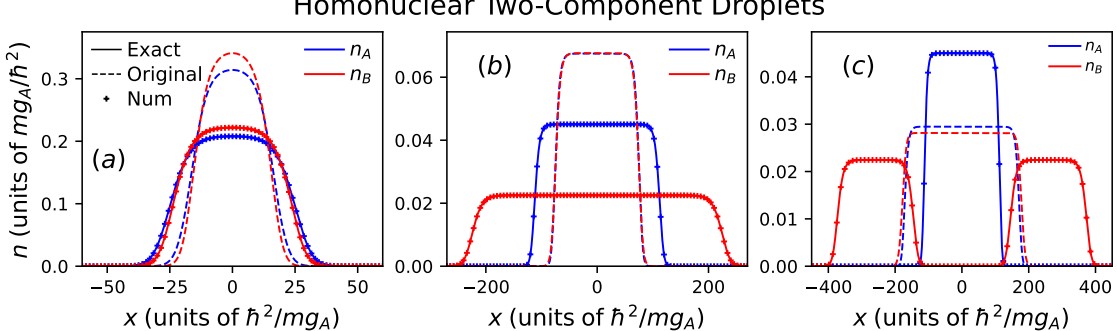

Figure 2: Ground state density configurations of a homonuclear two-component mixture for fixed $N_A = N_B \equiv N = 10$, $m_A = m_B \equiv m$, and $g_B = 0.5g_A$. Different intercomponent coupling strengths are considered ranging from (a) attractive $g_{AB} = -0.4g_A$, to (b) decoupled $g_{AB} = 0$, and (c) repulsive $g_{AB} = 0.4g_A$ cases. The outcome of the exact [Eq. (14)] and the original [Eq. (5)] eGPEs is illustrated as well as the one stemming from numerical integration of the sum of the LHY energy [Eq. (11)]. As for the balanced mixture, see Fig. 1, the original eGPEs overestimate the peak density of the droplets and underestimate their width. Interestingly, as long as the intercomponent interaction becomes repulsive [panel (c)], an early onset of the miscible-to-immiscible transition, coexisting with droplet configurations, appears to take place, and being solely captured by the exact eGPEs.

A close inspection of Fig. 1 reveals that the original eGPEs (Eq. (5)) are only accurate when $|g_{AB}| = g$. In contrast, deviating from this condition they significantly overestimate the droplet's saturation density and localization in all other interaction regimes, see Fig. 1(a)-(d). In particular, the difference in the saturation density among the two approaches becomes larger for intermediate intercomponent couplings lying in the interval $-g < g_{AB} < g$ as illustrated in Fig. 1(f). Even more, for $g_{AB} = -0.6$ the original eGPEs predict a Gaussian profile for the ground state density and they fail to capture the FT density profile of the droplet as predicted by the exact eGPEs, see Fig. 1(b). Additionally, as mentioned before, it is clear that the droplet solution exists throughout the MF stability regime within both approaches, exhibiting FT configurations (for sufficiently large particle number) irrespectively of the intercomponent interaction, $g_{AB}$, value [Fig. 1(a)-(d)]. Specifically, the droplet becomes progressively more delocalized and its saturation density decreases as $g_{AB}$ varies from $g_{AB} = -g$ to $g_{AB} = g$. Namely, the droplet saturation density (width) reduces (increases) sharply between the two limits, reaching a minimum value of $n_s(g_{AB} = g) = \frac{2mg}{9\hbar^2\pi^2} \approx 0.0225$ for the parameter values considered here, see also Fig. 1(f). As such, it could be that in practice, after a certain value of $g_{AB}$ the droplet would be dilute and extended enough appearing to have effectively disperse thus rendering challenging its experimental resolution by the corresponding imaging apparatus.

As a next step, we turn our attention to an intracomponent interaction imbalanced homonuclear ($m_A = m_B \equiv m$) mixture, with $g_A \neq g_B$ and fixed $g_B = 0.5g_A$, as well as $N_A = N_B \equiv N = 10$. The corresponding two-component droplet configurations within the exact [Eq. (14)] and original [Eq. (5)] eGPEs, but also the solutions obtained by calculating the LHY energy through direct numerical integration of the sum in Eq. (11) are visualized in Fig. 2 upon different variations of the $g_{AB}$ attraction.[2] For clarity, the ensuing solutions are marked by "Exact", "Original", and "Num" respectively. In general, it appears that as long as

---

[2]We remark that a larger $g_B$ repulsion results in more localized structures in both components, without any qualitative differences from the configurations depicted in Fig. 2 (not shown).

the intercomponent coupling is attractive (i.e. $g_{AB} < 0$) the two components prefer to maximize their spatial overlap [30, 61], while exhibiting similar density profiles as can be seen in Fig. 2(a). Moreover, tuning $g_{AB}$ towards weaker attractions, both components become less localized, until a transition from a Gaussian-type to a FT density profile takes place, compare in particular Fig. 2(a) and (b). Once more, as in the fully symmetric mixture, the original eGPEs predict appreciably more localized droplet densities alongside a higher FT value.

Additionally, approaching the weakly coupled intercomponent regime ($g_{AB} \approx 0$) the individual component droplet separation becomes more prominent, with the more strongly repulsive component exhibiting a more localized density profile and a higher saturation density [Fig. 2(b)]. Strikingly, this effect is absent within the original eGPEs, predicting instead a largely overlapping behavior for all intercomponent interactions shown in Fig. 2. Turning to repulsive intercomponent couplings, e.g. $g_{AB} = 0.4g_A$, we observe a phase separation of the droplet density between the two components, even though we are still operating well within the MF stability regime, see Fig. 2(c). A behavior that is completely absent within the original eGPEs. This phase separation can be understood from energetic arguments. Indeed, by ignoring the spatial overlap between the two components, it is possible to approximate the system shown in Fig. 2(c) by three distinct single-component droplets. Estimating their energy, in the homogeneous FT case, through minimization of the energy per particle (see also Eq. (12)) for each component in the absence of the other, results in $E_\sigma^{1\text{comp}} = -\frac{8m_\sigma^3 g_\sigma^3}{81\hbar^4\pi^4}L_\sigma$. Here, $L_\sigma$ designates the spatial extent of each individual droplet. With this approximation (setting $L_\sigma$ as the full-width-at-half-maximum for each structure) we find the total energy $E = E_A^{1\text{comp}} + 2E_B^{1\text{comp}} \approx -0.2832$ which is in excellent agreement with the numerically obtained energy $E = -0.2816$. The relatively small deviations stem from the (positive) contributions of the finite overlap and the kinetic terms. In particular, we can approximate the kinetic energy of each droplet as $K.E._\sigma = \int \frac{\hbar^2}{2m_\sigma}|\frac{\partial \Psi_\sigma}{\partial x}|^2 dx \approx 2\frac{\hbar^2}{2m_\sigma}|\frac{\sqrt{n_s}}{\xi_\sigma}|^2\xi_\sigma = \frac{8m_\sigma g_\sigma^2}{27\hbar^2\pi^2} = \frac{2g_\sigma}{3\pi}n_s$, where $\xi_\sigma$ is the healing length of the droplet and the factor of two stems from the two edges of the droplet. Interestingly, this suggests that the kinetic energy of the droplet scales as $\propto m_\sigma g_\sigma^2$ explaining why the more weakly interacting (and as we shall see below the lighter component) splits preferentially in the phase separated configurations, in order to reduce the systems kinetic energy. Clearly, an overlapping arrangement, with component B in a single FT configuration centered at $x = 0$ and length $L_B' \approx 2L_B$ would be associated with higher energy due to the repulsive intercomponent coupling. The energy contribution of the latter is $E_{AB} \approx g_{AB}n_A n_B L_{overlap}$, and it would be of order $\sim +0.1$ assuming similar saturation densities to those in Fig. 2(c) and $L_{overlap} \approx L_A \leq L_B'$. Hence, this hypothetical overlapping configuration would have a total energy of order $E' \approx E_A^{1\text{comp}} + E_B^{1\text{comp}}(L_\sigma = L_B') + E_{AB} = E + E_{AB} \sim -0.18$, which is significantly larger that the phase separated configuration.

Note that throughout this work, we focus on parity symmetric[3] density configurations, an assumption that pertains only to phase-separated structures. The droplet equations, however, support also parity asymmetric (or domain-wall like) configurations which, in the case of phase-separation, possess slightly lower energies (typically of the order of 0.3%) than the parity symmetric splitted configurations e.g. depicted in Fig. 2(c). Such parity asymmetric distributions while existing within the classical field approximation are not, in general, permitted within a many-body wavefunction perspective which in the absence of an explicitly symmetry breaking contribution in the Hamiltonian should retain the symmetry. In this context, the lowest energy parity symmetric configurations correspond to a good approximation to the reduced one-body density of the underlying atomic gas in the ground state, see also Appendix B for further details and the respective asymmetric density distributions. It is important

---

[3]Due to translation invariance, parity symmetry is defined with respect to the arbitrary center-of-mass of the mixture, here set to $x = 0$.

to emphasize that these asymmetric single domain-wall like configurations are intriguing on their own right and deserve further investigation, while they are promising for being exploited for unprecedented nucleation of topological excitations and hydrodynamic instabilities in the droplet realm. Finally, it is worth noting that in the presence of e.g. an external harmonic confinement, the components phase-separate at the standard miscibility threshold [23,73].

### 3.2 Heteronuclear two-component mixtures

Besides homonuclear mixtures, droplets can also be hosted in genuinely heteronuclear settings consisting of two different isotopes. Such systems have already been experimentally investigated in 3D deploying the isotopes $^{87}$Rb and $^{41}$K [45,74] or $^{87}$Rb and $^{23}$Na [75] but also in 1D with the former composition [47]. It is a known fact that the ensuing 3D eGPEs are far more complicated compared to the homonuclear ones due to the arguably complex form of the LHY contribution. However, thus far, in 1D (to the best of our knowledge) the extraction of the respective eGPEs remains elusive. Below, we elaborate on their construction and provide corresponding numerical results based on this framework.

Specifically, in the most general case of a heteronuclear mixture ($m_A \neq m_B$) the integral encompassing the LHY contribution represented by Eq. (11) cannot be calculated in a closed analytical form. An interesting limit arises for $\frac{g_B n_B}{g_A n_A} = \frac{m_B}{m_A}$ and $g_{AB} \approx 0$. Recall that the first of these conditions when $m_A = m_B$ reduces to the usual density fixing relation where the two-component eGPE framework reduces to a single-component one [20,53]. Retaining both (interaction) conditions, the LHY energy can be calculated analytically acquiring the closed form

$$\mathcal{E}_{LHY} = \frac{E_{LHY}}{L} = -\frac{m_A^2 + m_B^2}{2\hbar\pi}\left(\frac{g_A n_A}{m_A} + \frac{g_B n_B}{m_B}\right)^{\frac{3}{2}}. \tag{18}$$

However, this scenario is somewhat specialized. It requires the presence of appreciable experimental tunability in terms of Feshbach resonances that is not yet reached for these recently realized systems. As such, in what follows, we focus on general heteronuclear setups that do not satisfy the above-discussed conditions.

More concretely, we first and foremost numerically calculate the LHY energy density given by Eq. (11) and hence $\frac{\partial \mathcal{E}_{LHY}}{\partial n_\sigma}$ to be incorporated in the corresponding eGPEs featuring intercomponent mass imbalance, $m_A \neq m_B$. For the sake of comparison, we will also contrast the above-discussed ground state droplet densities with the ones obtained from either the original (Eq. (5)) or exact (Eq. (14)) eGPEs (exact only in the mass-balanced case, $m_A = m_B \equiv m$), by directly substituting the different masses of each component in the respective equation such that we can "artificially" emulate the heteronuclear setting. We mainly focus on moderate mass-imbalances with $m_B = 2m_A$ to unveil the basic droplet structures and afterwards comment on the impact of increasing imbalance.

For sufficiently strong intercomponent attraction, see Fig. 3(a), namely within the Gaussian droplet regime, the exact homonuclear eGPEs prediction is in excellent agreement with the fully general numerical approach. Deviations among the two approaches arise and progressively increase for decreasing intercomponent attraction, where the system transitions towards the FT region, see Fig. 3(b) where $g_{AB} = -0.4g_A$. In both cases intercomponent mixing is minor, i.e. the droplet profiles are close to one another, and becomes gradually more pronounced as we move to the noninteracting limit presented in Fig. 3(c) where the droplets maintain their FT structure. In contrast, for repulsive $g_{AB}$, the heteronuclear system prefers to phase separate into a central droplet occupied by the heavier component, while the lighter species fragments into two FT droplets on either side [Fig. 3(d)]. This preference of the lighter component to allocate into two different fragments is attributed to its reduced kinetic energy $\sim m_\sigma g_\sigma^2$ as compared to the corresponding one of the heavier component (see also the discussion at the

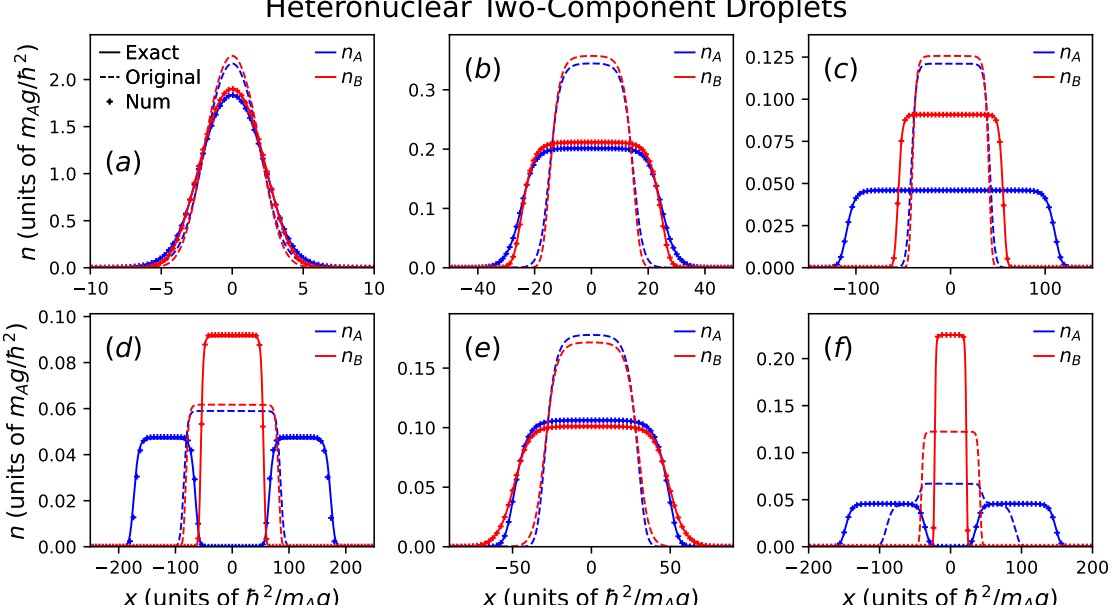

Figure 3: Ground state phases of heteronuclear droplet densities for (a)-(d) $m_B = 2m_A$, (e) $m_B = 0.5m_A$, and (f) $m_B = 5m_A$, whilst $N_A = N_B \equiv N = 10$, $g_A = g_B \equiv g$. The results are obtained within the original mass-balanced eGPEs [Eq. (5)] marked by "Original", the exact mass-balanced eGPEs given by Eq. (14) and indicated as "Exact", and the full numerically calculated heteronuclear eGPEs designated by "Num". In the first two cases, the mass of each component is plugged in the corresponding equation. The heteronuclear systems feature attractive (a) $g_{AB} = -0.8g$, and (b), (e) $g_{AB} = -0.4g$, (c) decoupled, $g_{AB} = 0$, and (d), (f) repulsive $g_{AB} = 0.4g$ intercomponent couplings. Interestingly, there is an excellent agreement between the ground state droplet densities predicted by the mass-balanced exact eG-PEs [Eq. (14)] and the fully general numerical approach. Minor deviations are observed for small mass differences and repulsive interactions which are, for instance, hardly visible in panel (d). Conversely, the original eGPEs [Eq. (5)] exhibit significant quantitative disagreement.

end of Sec. 3.1). In this case we observe a relatively small disagreement (hardly visible) with the fully general numerical approach, primarily at the edges of the droplet. As in the case of homonuclear mixtures we enforce parity symmetry, while asymmetric configurations are addressed in Appendix B.

On the other hand, the original eGPEs [Eq. (5)] completely miss this phase-separation behavior, predicting miscible droplets in the repulsive limit, while overestimating (underestimating) as before the droplet peak (width). For completeness, note that by employing a lighter species for component B (e.g. $m_B = 0.5m_A$) results in the same to the above-discussed phenomenology with the droplet configurations reversed while occupying overall larger length scales, compare for instance Fig. 3(b) and (e) with $m_B = 2m_A$ and $m_B = 0.5m_A$ respectively. In general, we find that both the exact (for $m_A = m_B \equiv m$) eGPEs [Eq. (14)] and the full numerical treatment of the heteronuclear mixture do not unveil fundamentally altered droplet configurations with respect to the mass-imbalance. As an example, for $m_B = 0.2m_A$, $0.5m_A$, $2m_A$, or $5m_A$ variations mainly the droplet localization is affected (not shown for brevity). Instead, the original eGPEs [Eq. (5)] exhibit high sensitivity to the mass ratio predicting significantly different behavior, see for instance Fig. 3(d), (f) with $m_B = 0.5m_A$ and $m_B = 5m_A$ where structural deformations are at play.

Concluding our investigation on the characterization of 1D two-component droplet setups, it is worth mentioning explicitly that neglecting the LHY correction and hence reducing our description to the standard GPEs the system displays the properties of a uniform gas. This in part stems from the absence of wave collapse in 1D. Indeed, all the droplet structures that we revealed as well as the ones to be discussed in the following Sections are inherently owed to beyond MF effects. This means that they represent a direct manifestation of the involvement of quantum fluctuations as accounted for by the perturbative LHY correction.

## 4  Three-component mixture

Here, we extend our considerations on the construction of the relevant eGPEs to three-component 1D droplet settings. For their realization, three different states of $^{39}$K could be potentially employed. The understanding and properties of such states are largely unexplored in all spatial dimensions [62, 63, 65, 67], while their 1D description through eGPEs taking the appropriate LHY correction is, to the best of our knowledge, missing. Following the standard assumptions, we only consider two-body contact interparticle interactions and restrict the Hamiltonian to include up to bilinear combinations of operators. It is then straightforward to obtain the Hamiltonian for the three-component mixture, by employing the corresponding field operators $(\hat{\Psi}_A(x), \hat{\Psi}_B(x), \hat{\Psi}_C(x))$ as was done in Eq. (6). Along these lines, the three-component Hamiltonian is readily found to be (see also Ref. [62] for the 3D case)

$$
\begin{aligned}
\hat{\mathcal{H}} = {}& \frac{g_A N_A^2}{2L} + \frac{g_B N_B^2}{2L} + \frac{g_C N_C^2}{2L} + \frac{g_{AB} N_A N_B}{L} + \frac{g_{AC} N_A N_C}{L} + \frac{g_{BC} N_B N_C}{L} \\
& + \sum_{k \neq 0} \Big[ \Big( \frac{\hbar^2 k^2}{2m_A} + g_A n_A \Big) \hat{a}_k^\dagger \hat{a}_k + \Big( \frac{\hbar^2 k^2}{2m_B} + g_B n_B \Big) \hat{b}_k^\dagger \hat{b}_k + \Big( \frac{\hbar^2 k^2}{2m_C} + g_C n_C \Big) \hat{c}_k^\dagger \hat{c}_k \Big] \\
& + \frac{1}{2} \sum_{k \neq 0} \Big[ g_A n_A \big( \hat{a}_k \hat{a}_{-k} + \hat{a}_k^\dagger \hat{a}_{-k}^\dagger \big) + g_B n_B \big( \hat{b}_k \hat{b}_{-k} + \hat{b}_k^\dagger \hat{b}_{-k}^\dagger \big) + g_C n_C \big( \hat{c}_k \hat{c}_{-k} + \hat{c}_k^\dagger \hat{c}_{-k}^\dagger \big) \Big] \\
& + g_{AB} \sqrt{n_A n_B} \sum_{k \neq 0} \big( \hat{a}_k \hat{b}_{-k} + \hat{a}_k^\dagger \hat{b}_{-k}^\dagger + \hat{a}_k \hat{b}_k^\dagger + \hat{a}_k^\dagger \hat{b}_k \big) \\
& + g_{AC} \sqrt{n_A n_C} \sum_{k \neq 0} \big( \hat{a}_k \hat{c}_{-k} + \hat{a}_k^\dagger \hat{c}_{-k}^\dagger + \hat{a}_k \hat{c}_k^\dagger + \hat{a}_k^\dagger \hat{c}_k \big) \\
& + g_{BC} \sqrt{n_B n_C} \sum_{k \neq 0} \big( \hat{b}_k \hat{c}_{-k} + \hat{b}_k^\dagger \hat{c}_{-k}^\dagger + \hat{b}_k \hat{c}_k^\dagger + \hat{b}_k^\dagger \hat{c}_k \big) + \mathcal{O}(\hat{a}_k^3).
\end{aligned}
\tag{19}
$$

Next, by defining the set of operators $\Phi^\dagger = (\hat{a}_k^\dagger, \hat{b}_k^\dagger, \hat{c}_k^\dagger, \hat{a}_{-k}, \hat{b}_{-k}, \hat{c}_{-k})$, which satisfy the bosonic commutation condition $\mathcal{M}_b = [\Phi, \Phi^\dagger] = \begin{pmatrix} I_3 & 0 \\ 0 & -I_3 \end{pmatrix}$, where $I_3$ is the $3 \times 3$ identity matrix, we can express the Hamiltonian of Eq. (19) in the following bilinear form

$$
\begin{aligned}
\hat{H} = {}& \sum_{k > 0} \Phi^\dagger H \Phi + \frac{g_A N_A^2}{2L} + \frac{g_B N_B^2}{2L} + \frac{g_C N_C^2}{2L} + \frac{g_{AB} N_A N_B}{L} + \frac{g_{AC} N_A N_C}{L} + \frac{g_{BC} N_B N_C}{L} \\
& - \sum_{k > 0} \sum_\sigma \Big( \frac{\hbar^2 k^2}{2m_\sigma} + g_\sigma n_\sigma \Big).
\end{aligned}
\tag{20}
$$

The respective Hamiltonian matrix ($H$), due to its involved form is provided in Appendix A. As it was shown in Sec. 2.2, this Hamiltonian can be diagonalized by solving its characteristic

equation (see also Eq. (3)) $\det(\mathcal{M}_b H - \omega_k I_6) = 0$, which yields the characteristic polynomial

$$
\begin{aligned}
\omega^6 &- (\epsilon_A^2 + \epsilon_B^2 + \epsilon_C^2)\omega^4 + \left[\epsilon_A^2\epsilon_B^2 + \epsilon_A^2\epsilon_C^2 + \epsilon_B^2\epsilon_C^2 - 4(J_{AB} + J_{AC} + J_{BC})\right]\omega^2 \\
&+ \left[-\epsilon_A^2\epsilon_B^2\epsilon_C^2 + 4(J_{AB}\epsilon_C^2 + J_{AC}\epsilon_B^2 + J_{BC}\epsilon_A^2) - 16\,\mathrm{sgn}(g_{AB}g_{AC}g_{BC})\sqrt{|J_{AB}J_{AC}J_{BC}|}\right] = 0.
\end{aligned}
\tag{21}
$$

As expected, it contains the single-component Bogoliubov energies $\epsilon_\sigma^2 = (\frac{\hbar^2 k^2}{2m_\sigma})^2 + \frac{\hbar^2 k^2}{m_\sigma}g_{\sigma\sigma}n_\sigma$ and $J_{\sigma\sigma'} = \frac{\hbar^2 k^2}{2m_\sigma}\frac{\hbar^2 k^2}{2m'_\sigma}g_{\sigma\sigma'}^2 n_\sigma n_{\sigma'}$ which represents the interspecies coupling contribution.

The roots $(\omega_1^2, \omega_2^2, \omega_3^2)$ of the characteristic polynomial can be calculated analytically, albeit they have a rather involved form in the general case. Accordingly, the ground state energy becomes

$$
\begin{aligned}
E_0 = &\frac{g_A N_A^2}{2L} + \frac{g_B N_B^2}{2L} + \frac{g_C N_C^2}{2L} + \frac{g_{AB} N_A N_B}{L} + \frac{g_{AC} N_A N_C}{L} + \frac{g_{BC} N_B N_C}{L} \\
&+ \sum_{k>0}\left(\omega_1 + \omega_2 + \omega_3 - \frac{\hbar^2 k^2}{2m_A} - \frac{\hbar^2 k^2}{2m_B} - \frac{\hbar^2 k^2}{2m_C} - g_A n_A - g_B n_B - g_C n_C\right).
\end{aligned}
\tag{22}
$$

Apparently, already at the MF level, the system is characterized by 12 independent variables spanning the parameter space $(g_{\sigma\sigma'}, n_\sigma, m_\sigma)$. This renders rather challenging the systematic exploration of ensuing phases and development of a unified understanding in the general case. Hence, it is more convenient and instructive (at least for the purpose of this work) to address only certain characteristic cases, both from the MF and even more so from the LHY perspective.

## 4.1 Fully symmetric mixture

The simplest reduction of the three-component system corresponds to that of a fully symmetric mixture, which can be retrieved by requiring $m_\sigma = m$, $g_{\sigma\sigma} = g$, $g_{\sigma\sigma'} = G$, and $n_\sigma = n$ for every $\sigma \neq \sigma'$. It is important to first examine this reduction since, despite its simplicity, it offers a useful benchmark with the respective single-component droplet system [31, 53], while deviating from it allows to build-up a systematic understanding of the more complex three-component setting. Employing the above-mentioned assumptions from Eq. (22) we can readily obtain the underlying ground state energy density

$$
\begin{aligned}
\mathcal{E} = \frac{E_0}{L} &= \frac{3g n^2}{2} + 3G n^2 + \frac{1}{L}\sum_{k>0}\left(\omega_1 + \omega_2 + \omega_3 - \frac{3\hbar^2 k^2}{2m} - 3g n\right) \\
&= 3\left(\frac{g}{2} + G\right)n^2 - \underbrace{\frac{2(mg^3 n^3)^{\frac{1}{2}}}{3\hbar\pi}\left[(1 + \frac{2G}{g})^{\frac{3}{2}} + 2(1 - \frac{G}{g})^{\frac{3}{2}}\right]}_{\mathcal{E}_{\mathrm{LHY}}^{3\mathrm{comp}}},
\end{aligned}
\tag{23}
$$

where $\omega_1 = \omega_2 = \sqrt{\epsilon^2 - 2J}$ and $\omega_3 = \sqrt{\epsilon^2 + 4J}$[4] with Bogoliubov energies $\epsilon^2 = (\frac{\hbar^2 k^2}{2m})^2 + \frac{\hbar^2 k^2}{m}g n$ and interspecies coupling contribution $J = \frac{\hbar^2 k^2}{2m}G n$. Finally, for convenience we define the dimensionless density independent parameter

$$
F(G/g) = \mathcal{E}_{\mathrm{LHY}}^{3\mathrm{comp}}/\sqrt{mg^3 n^3 \hbar^{-2}} = -\frac{2}{3\pi}\left[(1 + 2G/g)^{3/2} + 2(1 - G/g)^{3/2}\right] < 0.
$$

Note that as long as $-g/2 \leq G \leq g$ (which corresponds to the MF stability regime as discussed in Appendix A) the eigenvalues $\omega_i$ are real. Also, in line with the two-component case, the

---

[4]These are the solutions of the reduced characteristic polynomial equation

$$
\omega^6 - 3\epsilon^2\omega^4 + (3\epsilon^4 - 12J^2)\omega^2 + (-\epsilon^6 + 12\epsilon^2 J^2 - 16J^3) = 0.
$$

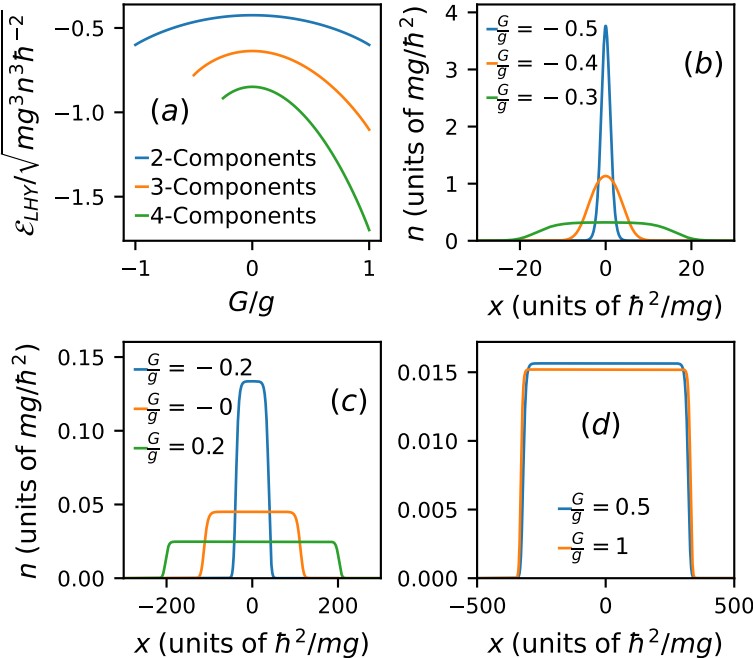

Figure 4: (a) The behavior of the LHY energy for the symmetric mixture stemming from a reduction of the genuine two-, three- and four-component settings as a function of the intercomponent interaction $G$. (b)-(d) Ground state densities of a fully symmetric three-component mixture with $N = 10$ for varying interaction $G$ (see legends). A transition from a Gaussian to a FT profile with increasingly larger spatial extent, upon reducing the attraction [panels (b), (c)] or increasing repulsion [panels (c), (d)] is evident.

symmetric three-component mixture reduces to an effectively single-component one. This effectively single-component system, however, depends on two interaction parameters, namely $g$ and $G$. It is inherently different from the corresponding single field reduction stemming from the two-component setup exclusively due to the respective LHY corrections (see the discussion below).

The LHY energy of this reduced system, again remains $\sim n^{3/2}$ throughout the MF stability regime $G \in [-g/2, g]$. Moreover, the density independent proportionality factor ranges from $F(G/g) = -\sqrt{6}/\pi$ for $G = -g/2$, to $F(G/g) = -2\sqrt{3}/\pi$ for $G = g$, while it takes its maximum $F(G/g) = -2/\pi$ value at $G = 0$, as shown in Fig. 4(a). Interestingly, the strength of quantum fluctuations appears to be larger close to the immiscibility threshold $G = g$ as compared to the one close to the droplet regime $G = -g/2$, as evidenced by the more strongly attractive LHY energy (see e.g. Fig. 4(a) and Eq. (23)). This is in contrast to the two-component mixture where quantum fluctuations contribute equally at the two limits, see in particular Fig. 1(e) and Eq. (13). We attribute this behavior of the three-component system to the aforementioned "asymmetric" MF stability regime (see also Appendix A for a more elaborated discussion on its asymmetric nature).

On the other hand, the MF interaction term vanishes at the droplet threshold $G \approx -g/2$, while it takes its maximum value at the immiscibility threshold $g = G$. Hence, it is not *a-priori* obvious how important the role of the LHY correction becomes at each limit (e.g. from Eq. (23)) and one has to solve the corresponding eGPE, as we shall do below. Finally, it is interesting to note that in the fully symmetric case the corresponding LHY energy of the original three-component mixture is always smaller than the one of the reduced two-component

system, namely it holds that $\left|\mathcal{E}_{\text{LHY}}^{\text{3comp}}\right| \geq \frac{3}{2}\left|\mathcal{E}_{\text{LHY}}^{\text{2comp}}\right|$ with the equality being valid for $G = 0$. As such, it appears that the effect of quantum fluctuations is enhanced in the case of a three-component mixture, even beyond the additive effect owing to the additional component, due to the interspecies coupling. Instead, for the MF energy it holds that $\mathcal{E}_{\text{MF}}^{\text{3comp}}/3 = \mathcal{E}_{\text{MF}}^{\text{2comp}}/2$, i.e. the ratio of the MF energy divided by the number of components remains constant for any number of components in the symmetric case.

To obtain the effective single-component eGPE equation emanating from the fully symmetric three-component mixture we evaluate the Euler-Lagrange equations (from Eq. (23)) with respect to $n$. This standard procedure leads to the symmetric eGPE:

$$3i\hbar\frac{\partial\Psi}{\partial t} = -3\frac{\hbar^2}{2m}\frac{\partial^2\Psi}{\partial x^2} + 3(g + 2G)|\Psi|^2\Psi + F(G/g)\frac{3\sqrt{mg^3}}{2\hbar}|\Psi|\Psi. \tag{24}$$

Similarly to the case of a symmetric two-component mixture, this symmetric eGPE accepts a droplet solution with saturation density $n_s = F^2(G/g)\frac{mg^3}{9\hbar^2(2G+g)^2}$, energy density $\mathcal{E}^{3comp}(n = n_s) = -\frac{3}{2}(g + 2G)n_s^2$, chemical potential $\mu_s = -\frac{3}{2}(g + 2G)n_s$, and healing length $\xi = \sqrt{-\frac{3\hbar^2}{2m\mu_s}} = \sqrt{\frac{9\hbar^4(g+2G)}{m^2g^3F^2(G/g)}} = \sqrt{\frac{\hbar^2}{m(g+2G)n_s}}$. Again, in accordance with the two-component symmetric mixture, the ground state density of the symmetric three-component system transits from a Gaussian-type profile (for $G \approx -1/2g$) to a FT one for decreasing intercomponent attraction as captured by the $G$ coefficient. This behavior is explicitly illustrated in Fig. 4(b) for a fixed atom number.[5] Additionally, the ground state density maintains its FT character throughout the repulsive interaction regime, $G > 0$, see Fig. 4(c) and (d). The saturation density [width] decreases [increases] by over an order of magnitude as $G \approx g$, eventually acquiring the minimum saturation density $n_s(G = g) = \frac{4mg}{27\pi^2\hbar^2} \approx 0.015$ deep in the repulsive interaction regime as can be seen in Fig. 4(d).

## 4.2  Two identical components

As it was argued above, already in the limiting case of a fully symmetric three-component mixture evident differences arise (concerning, for instance, the droplet saturation density, and the LHY correction) when compared to the corresponding reduced two-component system. However, by construction three-component mixtures offer additional possibilities for intercomponent asymmetry, that could lead to intriguing phenomena and phases that can not be captured within the two-component setting. A step forward to delve more into the three-component properties is to consider a system where two of the components are identical (i.e. symmetric), while the third one is different. To be concrete, this situation is described by $m_A = m_B \equiv m$, $g_A = g_B \equiv g$, and $g_{AC} = g_{BC} \equiv G_C$, while $g_{AB}$ and $g_C$ are unrestricted parameters, see also Ref. [62] for the 3D case.

In this scenario, it can be found that the MF stability requires the following conditions to be fulfilled: i) $g, g_C > 0$, ii) $gg_C > G_C^2$, and $g > |g_{AB}|$ as well as iii) $G_C^2 < \frac{g_C}{2}(g + g_{AB})$, see also Appendix A for further details. We remark that the same conditions hold for the respective 3D system as reported in Ref. [62], which is to be expected because the homogeneous MF energy is independent of the dimensionality. It is also important to emphasize that the last (iii) condition is more restrictive than the second (ii) one since $g_C(g + g_{AB})/2 \leq gg_C$ if $|g_{AB}| < g$. As a consequence, it is possible for each pair of subsystems to be stable, while the overall system is outside the MF stability region, a behavior that equally holds in 3D [62]. For simplicity, here, we focus on interaction intervals where each subsystem and the overall mixture reside within the MF stability region.

---

[5]Similarly, for fixed interactions the FT transition occurs for increasing particle number (e.g. at $N \geq 30$ for $G = -0.4g$).

Further assuming that $m_C = m$ (for convenience), we find that the LHY energy density for the general three-component system in Eq. (22) takes the form

$$\mathcal{E}_{\text{LHY}} = -\frac{\sqrt{2m}}{6\hbar\pi}\left((2(g - g_{AB})n)^{\frac{3}{2}} + (W + Q)^{\frac{3}{2}} + (W - Q)^{\frac{3}{2}}\right), \tag{25}$$

with the parameters

$$Q = \left((g + g_{AB})^2 n^2 - 2(g + g_{AB})g_C n n_C + 8G_C^2 n n_C + g_C^2 n_C^2\right)^{\frac{1}{2}}, \tag{26a}$$

$$W = gn + g_{AB}n + g_C n_C. \tag{26b}$$

The LHY energy density is real and negative as long as $g > |g_{AB}|$ and $W - Q > 0 \rightarrow 2G_C^2 < g_C(g + g_{AB})$, which is exactly the MF stability regime, while it becomes complex otherwise. Moreover, by removing the third component namely setting $G_C = g_C = n_C = 0$, it holds that $Q = W = (g + g_{AB})n$ and the LHY energy density becomes $\mathcal{E}_{\text{LHY}} = \frac{-2\sqrt{m}(gn)^{3/2}}{3\hbar\pi}[(1 - g_{AB}/g)^{3/2} + (1 + g_{AB}/g)^{3/2}]$. This is exactly the LHY energy density of the symmetric two-component mixture in the MF stability regime (see Eq. (12) for $g_A = g_B \equiv g$ and $n_A = n_B \equiv n$).

As in Sec. 3.1, we can derive the LHY terms of the effective two-component eGPEs describing the system by calculating their partial derivatives

$$\frac{\partial \mathcal{E}_{\text{LHY}}}{\partial n} = -\frac{\sqrt{2m}}{4\pi h}\left(\left(g + g_{AB} + \frac{Z}{Q}\right)\sqrt{W - Q} + \left(g + g_{AB} - \frac{Z}{Q}\right)\sqrt{W + Q} + 2\sqrt{2}(g - g_{AB})^{\frac{3}{2}}\sqrt{n}\right), \tag{27a}$$

$$\frac{\partial \mathcal{E}_{\text{LHY}}}{\partial n_C} = -\frac{\sqrt{2m}}{4\pi h}\left(\left(g_C + \frac{P}{Q}\right)\sqrt{W - Q} + \left(g_C - \frac{P}{Q}\right)\sqrt{W + Q}\right), \tag{27b}$$

where

$$Z = -\frac{1}{2}\frac{\partial Q^2}{\partial n} = -g^2 n - 2g g_{AB}n + g g_C n_C - g_{AB}^2 n + g_{AB}g_C n_C - 4G_C^2 n_C, \tag{28a}$$

$$P = -\frac{1}{2}\frac{\partial Q^2}{\partial n_C} = g g_C n + g_{AB}g_C n - 4G_C^2 n - g_C^2 n_C. \tag{28b}$$

These LHY terms are incorporated in the corresponding eGPEs, which by additionally requiring $\Psi_A = \Psi_B \equiv \Psi$, read

$$2i\hbar\frac{\partial \Psi}{\partial t} = -\frac{\hbar^2}{m}\frac{\partial^2 \Psi}{\partial x^2} + \left(2(g + g_{AB})|\Psi|^2 + 2G_C|\Psi_C|^2 + \frac{\partial \mathcal{E}_{\text{LHY}}}{\partial n}\right)\Psi, \tag{29a}$$

$$i\hbar\frac{\partial \Psi_C}{\partial t} = -\frac{\hbar^2}{2m}\frac{\partial^2 \Psi_C}{\partial x^2} + \left(g_C|\Psi_C|^2 + 2G_C|\Psi|^2 + \frac{\partial \mathcal{E}_{\text{LHY}}}{\partial n_C}\right)\Psi_C. \tag{29b}$$

A few characteristic examples of the resulting ground state droplet configurations are presented in Fig. 5, with fixed $g_C = g$, and $g_{AB} = -0.5g$. Specifically, the case of $G_C = -0.5g$ refers to the respective LHY-fluid limit (attained in general here for $G_C = -g/2$ due to MF stability requirements) of the fully symmetric mixture where MF interactions cancel out. Here, the individual components become identical and possess a Gaussian type distribution, see Fig. 5(a). As we decrease the intercomponent attraction, the system transitions to a mixed state where the symmetric components $(A, B)$ exhibit a FT configuration and the third one $(C)$ becomes significantly distorted and delocalizes. Particularly, there is a FT droplet segment lying within the symmetric components and the remaining atoms of the third component reside outside this region in a self-bound state as shown in Fig. 5(b), (c). However, by switching-off the intercomponent interactions among the symmetric components and the third one, i.e. reaching the decoupling limit with $G_C = 0$, the subsystems feature two independent droplet structures [Fig. 5(d)].

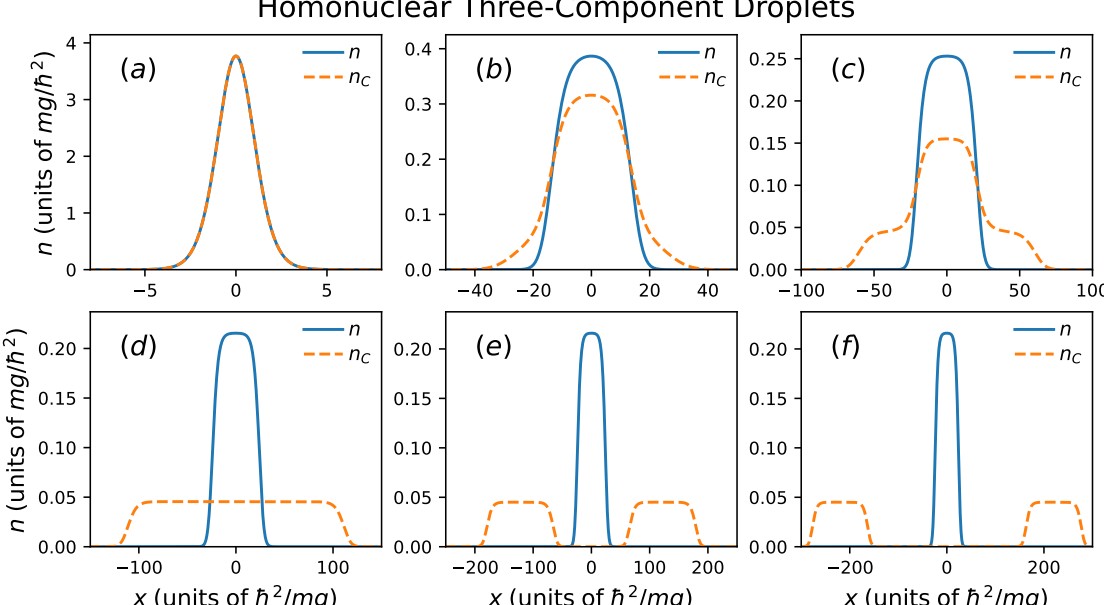

Figure 5: Droplet density configurations arising in a homonuclear three-component mixture for varying interspecies couplings $g_{AC} = g_{BC} \equiv G_C$. The latter range from the attractive (a) $G_C = -0.5g$, (b) $G_C = -0.2g$, (c) $G_C = -0.1g$, to (d) the decoupled $G_C = 0$, and (e) the repulsive $G_C = 0.1g$, (f) $G_C = 0.4g$ interaction regimes, thus covering the available parametric space. As it can be seen, mixed phases exhibiting a structurally deformed third component having both a droplet core and self-bound atoms at the tails [panel (c)] for weak intercomponent attractions or a configuration of two fully distinct droplets [panel (d)] in the decoupled regime occur. Also, an early onset of the miscible-to-immiscible transition, coexisting with droplet configurations, similarly to the two-component mixture takes place for repulsive intercomponent couplings. In all cases, the remaining system parameters are held fixed and in particular correspond to $N_A = N_B = N_C = 10$, $m_A = m_B = m_C \equiv m$, $g_A = g_B = g_C \equiv g$, and $g_{AB} = -0.5g$.

On the other hand, turning to the repulsive side of the MF stability regime (e.g. for $G_C = 0.1g$, or $G_C = 0.4g$), it turns out that the third component progressively separates from the symmetric two-component droplet and accommodates two smaller sized droplets on either side of the symmetric two-component droplet which remains localized at the center [Fig. 5(e), (f)] (see also Appendix B for the parity asymmetric configurations). Finally, it is important to emphasize that the above-described investigation is far from being exhaustive for the three-component system. A more detailed examination of the emerging droplet states and mixed phases of matter e.g. by varying $N_C$, $g_C$, $g_{AB}$, and $G_C$ across the MF stability regime is certainly an interesting direction to be pursued in future studies.

## 5 Four-component mixture

As can be deduced from the above analysis, in order to obtain the ground state energy of the quasi-particle vacuum one needs to diagonalize the Hamiltonian matrix which has $2 \times n$ dimensions, with $n$ representing the number of bosonic species. Due to the symmetries of the Hamiltonian (e.g. being a real Hermitian matrix) the resulting characteristic equation is

a polynomial of degree $n$, with respect to the eigenvalues $\omega^2$. Therefore, the four-component mixture is the most complex system for which the characteristic equation can be analytically solved and hence extract its LHY energy. For higher-component settings the characteristic equation can only be solved numerically in the general case. As such, for completeness, we present below the LHY energy for the four-component mixture, even though such a setup would be arguably more challenging to be experimentally realized.

The process for deriving the energy of the four-component mixture is identical to the ones of the two- (Sec. 3) and three- (Sec. 4) component ones described above. For brevity, we provide directly the resulting characteristic polynomial

$$\omega^8 + \alpha\omega^6 + \beta\omega^4 + \gamma\omega^2 + \delta = 0, \tag{30}$$

containing the rather involved coefficients

$$\alpha = -\sum_{\sigma}\epsilon_{\sigma}^2, \tag{31a}$$

$$\beta = \sum_{\sigma=A}^{D}\sum_{\sigma'>\sigma}^{D}\left(\epsilon_{\sigma}^2\epsilon_{\sigma'}^2 - 4J_{\sigma\sigma'}\right), \tag{31b}$$

$$\gamma = \sum_{\sigma}\left[-\epsilon_{\sigma}^2\left[\sum_{\sigma'>\sigma}^{D}\sum_{\sigma''>\sigma'}^{D}\epsilon_{\sigma'}^2\epsilon_{\sigma''}^2 + 4\sum_{\sigma'\neq\sigma}^{D}\sum_{\sigma''>\sigma',\sigma''\neq\sigma}^{D}J_{\sigma'\sigma''}\right]\right. \tag{31c}$$
$$\left.+ 16\prod_{\sigma'\neq\sigma,\sigma'<\sigma''}\text{sgn}(g_{\sigma'\sigma''})\sqrt{J_{\sigma'\sigma''}}\right],$$

$$\delta = \prod_{\sigma}\epsilon_{\sigma}^2 - 4\sum_{\mathcal{P}}\left[(\epsilon_{\sigma_1}^2\epsilon_{\sigma_2}^2 - 4J_{\sigma_1\sigma_2})J_{\sigma_3\sigma_4}\right] + 16\sum_{\sigma}\epsilon_{\sigma}^2\prod_{\sigma_{i,j}\neq\sigma,\sigma_i<\sigma_j}\frac{\sqrt{|J_{\sigma_i\sigma_j}|}}{\text{sgn}(g_{\sigma_i\sigma_j})} \tag{31d}$$
$$- 32\frac{\sqrt{|J_{AC}J_{AD}J_{BC}J_{BD}|}}{\text{sgn}(g_{AC}g_{AD}g_{BC}g_{BD})} - 32\frac{\sqrt{|J_{AB}J_{AD}J_{BC}J_{CD}|}}{\text{sgn}(g_{AB}g_{AD}g_{BC}g_{CD})} - 32\frac{\sqrt{|J_{AB}J_{AC}J_{BD}J_{CD}|}}{\text{sgn}(g_{AB}g_{AC}g_{BD}g_{CD})},$$

where $\epsilon_{\sigma}^2 = (\frac{\hbar^2k^2}{2m_{\sigma}})^2 + \frac{\hbar^2k^2}{m_{\sigma}}g_{\sigma\sigma}n_{\sigma}$ are the single component Bogoliubov energies and $J_{\sigma\sigma'} = \frac{\hbar^2k^2}{2m_{\sigma}}\frac{\hbar^2k^2}{2m'_{\sigma}}g_{\sigma\sigma'}^2 n_{\sigma}n_{\sigma'}$ the interspecies coupling contributions. The notation $\mathcal{P}$ in the sums refers to all possible permutations of $\sigma_1, \sigma_2, \sigma_3, \sigma_4$ such that $\sigma_1 < \sigma_2$, and $\sigma_3 < \sigma_4$ and thus index duplication is avoided.

The solutions ($\omega_i$) of the characteristic polynomial equation can be calculated exactly and the resulting ground state energy for the four-component quasi-particle vacuum reads

$$E_0 = \sum_{\sigma}\frac{g_{\sigma\sigma}N_{\sigma}^2}{2L} + \frac{1}{2}\sum_{\sigma\neq\sigma'}\frac{g_{\sigma\sigma'}N_{\sigma}N_{\sigma}'}{L} + \sum_{k>0}\left[\sum_{i=1}^{4}\omega_i - \sum_{\sigma}\left(\frac{\hbar^2k^2}{2m_{\sigma}} + g_{\sigma\sigma}n_{\sigma}\right)\right]. \tag{32}$$

It is apparent that for this setup the parametric space as defined by the different intra- and inter-component coupling combinations, the atom number and mass of each component becomes exceedingly large. While a systematic study of this system is interesting on its own right, in what follows we restrict ourselves to the relatively simpler case of the symmetric mixture.

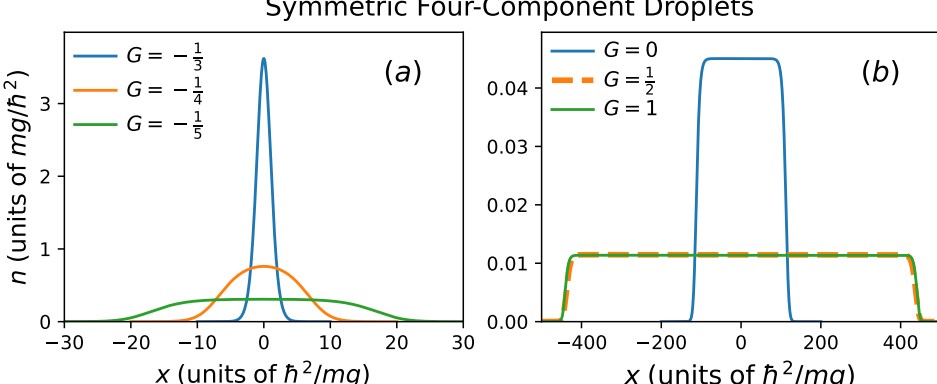

Figure 6: Ground state droplet densities within a fully symmetric four-component mixture with different intercomponent couplings $G$ (see legend) and fixed $N = 10$. The transition from a Gaussian-like to a FT profile takes place, upon reducing attraction [panel (a)] while for increasing repulsion [panel (b)], the droplets maintain their FT structure and exhibit increasingly larger spatial widths.

## 5.1 Symmetric four-component mixture

Let us consider a fully symmetric four-component mixture characterized by $N_\sigma \equiv N$, $g_{\sigma\sigma} \equiv g$, and $g_{\sigma\sigma'} \equiv G_4$. Here, the MF energy (first two sums of Eq. (32)) reduces to $\mathcal{E}_{\mathrm{MF}}^{4\mathrm{comp}} = (2g + 6G_4)n^2$, whilst the MF stability regime is given by $-g/3 < G_4 < g$, ensuring the stability of each subsystem but also of the full four component mixture. Accordingly, the roots of the characteristic polynomial of Eq. (30) simplify to $\omega_{1,2,3}^2 = \epsilon^2 - G_4\frac{\hbar^2 k^2}{m}n$, and $\omega_4^2 = \epsilon^2 + 3G_4\frac{\hbar^2 k^2}{m}n$ and the LHY energy density becomes

$$\mathcal{E}_{\mathrm{LHY}}^{4\mathrm{comp}} = -\frac{2\sqrt{mg^3 n^3}}{\pi\hbar}\left(\left(1 - \frac{G_4}{g}\right)^{\frac{3}{2}} + \frac{1}{3}\left(1 + 3\frac{G_4}{g}\right)^{\frac{3}{2}}\right). \tag{33}$$

As in the case of the symmetric two-component mixture [Sec. 3.1], the emergent droplet solution can be shown to have saturation density $n_s = \frac{mg^3}{4\pi^2\hbar^2(3G_4+g)^2}\left((1-\frac{G_4}{g})^{3/2} + \frac{1}{3}(1+3\frac{G_4}{g})^{3/2}\right)$, energy density $\mathcal{E}^{4\mathrm{comp}}(n=n_s) = -2(g+3G_4)n_s^2$, chemical potential $\mu_s = -2(g+3G_4)n_s$, and healing length $\xi = \sqrt{-\frac{\hbar^2}{2m\mu_s}} = \sqrt{\frac{\hbar^2}{4m(g+3G_4)n_s}}$. Moreover, similarly to the three-component setup, the LHY energy of the fully symmetric four-component mixture is always smaller than twice the LHY energy of the corresponding symmetric two-component system, namely $\mathcal{E}_{\mathrm{LHY}}^{4\mathrm{comp}} \leq 2\mathcal{E}_{\mathrm{LHY}}^{2\mathrm{comp}}$ with the equality being valid for $G_4 = 0$. This implies the presence of enhanced quantum fluctuations in the symmetric four-component system, see also Fig. 4(a), beyond an additive contribution. On the other hand, for the MF energies it holds that $\mathcal{E}_{\mathrm{MF}}^{4\mathrm{comp}} = 2\mathcal{E}_{\mathrm{MF}}^{2\mathrm{comp}}$. Representative ground state droplet profiles of the reduced four-component setting are illustrated in Fig. 6 for different intercomponent couplings ($G_4$) and fixed intracomponent ones. In line with the behavior of the droplets arising in lower-component mixtures also here a deformation from a Gaussian to a FT density profile occurs for decreasing intercomponent attractions as depicted in Fig. 6(a). Finally, at the repulsive side, increasing the intercomponent interaction across the MF stability regime results in quantum droplet structures of significantly larger spatial width, as shown in Fig. 6(b).

# 6 Conclusions and future directions

We derived, in a systematic manner, the first-order beyond mean-field LHY quantum correction and associated eGPEs for two-, three- and four-component 1D bosonic mixtures featuring short-range contact interactions. The four-component system represents the highest multicomponent setting for which the LHY energy can be analytically derived in the general case. In all cases, whenever the complexity of the system allows corresponding closed form expressions are provided. Importantly, our treatment relies only on the assumption of a dilute gas such that the perturbative Bogoliubov treatment is valid. Hence, the LHY energies and the ensuing eGPEs obtained are reliable throughout the MF stability regime, namely from the attractive to the repulsive intercomponent interaction regimes in contrast to previous treatments.

Focusing on the two-component mixture, we extracted closed form solutions for the LHY energy and constructed the corresponding exact coupled system of eGPEs for homonuclear settings but also provided numerical results for heteronuclear 1D systems. Specifically, by tuning the intercomponent interactions towards smaller attractions, we retrieve the structural deformation of the droplets from Gaussian to FT density configurations. However, it is explicated that the previously discussed original two-component LHY treatment reveals quantitative deviations as compared to our exact eGPE results and can be obtained as a limiting case from the exact expressions presented herein. These deviations are reflected by an increased saturation droplet density and decreasing width. Turning to repulsive interactions, we identify even more significant qualitative differences with the original LHY treatment, especially for heteronuclear settings featuring intercomponent mass-imbalance. In particular, self-bound structures with decreasing saturation densities are found for larger intercomponent repulsions, while an early onset of phase-separation occurs for both homonuclear and heteronuclear mixtures. The numerically computed heteronuclear droplet states are found to be in good agreement with the results of the exact eGPEs for homonuclear settings upon direct substitution of the different masses. Meanwhile, the original eGPEs fail to adequately capture the droplet densities.

Additionally, exact closed form expressions of the LHY energies and the eGPEs are discussed for three-component mixtures especially in the cases of i) a fully-symmetric system and ii) two identical components coupled to a third independent one. It is argued that the LHY energy is not additive to the number of components and it is in general larger for three-component setups as compared to two-component ones. A plethora of droplet phases exist, ranging from the underlying LHY-fluid limit, to Gaussian miscible droplet components as well as to mixed states for decreasing attraction. These mixed states refer to FT configurations for the symmetric components and a delocalized droplet distribution for the third one. The latter assembles in a FT droplet segment within the symmetric components and the remaining atoms lying beyond the intercomponent overlap region are in a self-bound state. Instead, for repulsive interactions the third component phase separates from the symmetric two-component droplet and exhibits two smaller sized droplets on either side. Finally, a closed LHY form is also provided for the fully symmetric four-component mixture. Here, the transition from Gaussian to FT droplets for decreasing attraction and afterwards to large spatial FT structures upon crossing to the repulsive intercomponent regime is observed.

There are several intriguing pathways to be followed in the future based on our work which provides the first steps towards exploring ground state multicomponent droplet configurations. A straightforward extension is to systematically study the transition towards phase-separation in heteronuclear mass-imbalanced two-component mixtures and unveil the origin of this mechanism. Similarly, the three- and four-component mixtures offer completely unexplored possibilities to create exotic droplet configurations. These include, for instance, miscible spatially deformed droplet structures, and importantly mixed droplet phases thereof as well as droplet-soliton [76] or droplet-vortex [36] coexistence. The interpolation of three-component droplet

states to the few-body regime using sophisticated numerical schemes is also of particular interest [77–79]. On the other hand, dynamical crossing of these droplet phases is expected to reveal insights into their collective excitations and the spontaneous generation of nonlinear defects or delocalized structures such as shock-waves [80, 81]. Along these lines, emulating the time-of-flight measurement process of multicomponent droplet states in order to testify their self-bound nature is certainly worth pursuing. Finally, extending our treatment to 3D geometries in order to investigate the impact of the LHY term beyond the MF stability edge as well as in the presence of external potentials are very relevant open questions to address.

## Acknowledgments

The authors thank P. G. Kevrekidis, G. Bougas and G.C. Katsimiga for extensive discussions on the topic of droplets.

**Funding information** S.I.M acknowledges support from the University of Missouri Science and Technology, Department of Physics, Startup fund.

## A Details on the stability conditions of the three-component mixture

The Hamiltonian matrix for the three-component mixture used in Eq. (20) of the main text takes the form

$$
H = \begin{pmatrix}
\frac{\hbar^2 k^2}{2m_A} + g_A n_A & g_{AB}\sqrt{n_A n_B} & g_{AC}\sqrt{n_A n_C} & g_A n_A & g_{AB}\sqrt{n_A n_B} & g_{AC}\sqrt{n_A n_C} \\
g_{AB}\sqrt{n_A n_B} & \frac{\hbar^2 k^2}{2m_B} + g_B n_B & g_{BC}\sqrt{n_B n_C} & g_{AB}\sqrt{n_A n_B} & g_B n_B & g_{BC}\sqrt{n_B n_C} \\
g_{AC}\sqrt{n_A n_C} & g_{BC}\sqrt{n_B n_C} & \frac{\hbar^2 k^2}{2m_C} + g_C n_A & g_{AC}\sqrt{n_A n_C} & g_{BC}\sqrt{n_B n_C} & g_C n_C \\
g_A n_A & g_{AB}\sqrt{n_A n_B} & g_{AC}\sqrt{n_A n_C} & \frac{\hbar^2 k^2}{2m_A} + g_A n_A & g_{AB}\sqrt{n_A n_B} & g_{AC}\sqrt{n_A n_C} \\
g_{AB}\sqrt{n_A n_B} & g_B n_B & g_{BC}\sqrt{n_B n_C} & g_{AB}\sqrt{n_A n_B} & \frac{\hbar^2 k^2}{2m_B} + g_B n_B & g_{BC}\sqrt{n_B n_C} \\
g_{AC}\sqrt{n_A n_C} & g_{BC}\sqrt{n_B n_C} & g_C n_C & g_{AC}\sqrt{n_A n_C} & g_{BC}\sqrt{n_B n_C} & \frac{\hbar^2 k^2}{2m_C} + g_C n_A
\end{pmatrix}.
$$

Before exploring the LHY term in more detail, it is important to determine the MF stability regime. In general, the stability condition is given by demanding that the Hessian matrix ($Hess$) in terms of derivatives with respect to the densities ($Hess_{\sigma,\sigma'} = \frac{\partial^2 \mathcal{E}_{\mathrm{MF}}}{\partial n_\sigma \partial n_{\sigma'}}$) has only positive eigenvalues [5]. Namely, the Hessian matrix in our case reads

$$
Hess = \begin{pmatrix}
g_A & g_{AB} & g_{AC} \\
g_{AB} & g_B & g_{BC} \\
g_{AC} & g_{BC} & g_C
\end{pmatrix},
\tag{A.1}
$$

with eigenvalues dictated by the roots of the characteristic polynomial

$$
f(x) = x^3 - x^2 \sum_\sigma g_{\sigma\sigma} + x \sum_{\sigma<\sigma'} \Delta g_{\sigma\sigma'} + C.
\tag{A.2}
$$

In this expression, $\Delta g_{\sigma\sigma'} = g_\sigma g_{\sigma'} - g_{\sigma\sigma'}^2$ and $C = g_A g_{BC}^2 + g_B g_{AC}^2 + g_C g_{AB}^2 - g_A g_B g_C - 2 g_{AB} g_{AC} g_{BC}$. Note that the stability of the individual components requires $\Delta g_{\sigma\sigma'} > 0$ (and $g_{\sigma\sigma} > 0$) for all

components. Then, MF stability requires

$$
\begin{aligned}
[1.] \quad & g_{\sigma\sigma} > 0, \\
[2.] \quad & \Delta g_{\sigma\sigma'} > 0, \\
[3.] \quad & C < 0, \\
[4.] \quad & \frac{1}{3}\sum_{\sigma} g_{\sigma\sigma} - \sqrt{\Delta} > 0, \quad \text{where } \Delta = \sum_{\sigma} g_{\sigma\sigma}^2 + \sum_{\sigma\neq\sigma'}(g_{\sigma\sigma'}^2 - \Delta g_{\sigma\sigma'}/2), \\
[5.] \quad & f\Big(\frac{\sum_{\sigma} g_{\sigma\sigma} - \sqrt{\Delta}}{3}\Big) \geq 0, \quad \text{and } f\Big(\frac{\sum_{\sigma} g_{\sigma\sigma} + \sqrt{\Delta}}{3}\Big) \leq 0.
\end{aligned}
\tag{A.3}
$$

In particular, the last two conditions guarantee the existence of three real solutions. However, the Hessian matrix (Eq. (A.1)) is a real and symmetric (normal) matrix, and hence it is guaranteed to be diagonalizable. As such, the characteristic polynomial (Eq. (A.2)) is already guaranteed to have three real roots by construction and we do not need to check the (rather complicated) conditions [5.] explicitly. Finally, condition [4.] is automatically satisfied as long as condition [2.] holds.

In the case of two symmetric components (i.e., $g_A = g_B = g$) coupled to the third component with the same interaction strength (namely $g_{AC} = g_{BC} = G_C$) the characteristic polynomial of the Hessian matrix becomes

$$
f(x) = x^3 - x^2(2g + g_C) + x(g^2 - 2G_C^2 - 2gg_C - g_{AB}^2) + [2G_C^2 g - 2G_C^2 g_{AB} - g^2 g_C + g_{AB}^2 g_C]. \tag{A.4}
$$

Therefore, MF stability for these three-component mixtures requires

$$
\begin{aligned}
[1.] \quad & g > 0, \quad \text{and} \quad g_C > 0, \\
[2.] \quad & |g_{AB}| > g, \quad \text{and} \quad gg_C > G_C^2, \\
[3.] \quad & G_C^2 < \frac{g_C}{2}(g + g_{AB}).
\end{aligned}
\tag{A.5}
$$

For the case of the fully symmetric mixture i.e. $g_A = g_B = g_C = g$ and $g_{AB} = g_{AC} = g_{BC}$, the system is significantly simplified and the MF stability requires $-g/2 < G < g$. Evidently, the MF stability window becomes reduced compared to the respective two-component one, given by $-g < g_{AB} < g$ as discussed in Sec. 3.1, and asymmetric with respect to the sign of the inter-species interaction strength. An intuitive explanation for this phenomenology is the following. Consider a mixture consisting of $\mathcal{N}$ identical components with $N$ particles per component, featuring intracomponent interactions $g_\sigma = g$ and intercomponent ones $g_{\sigma\sigma'} = G$, for all $\sigma$, $\sigma'$. Ignoring any complications stemming from the exchange statistics and distinguishability, a particle of species $\sigma$ may scatter from $N-1$ $\sigma$-species particles with strength $g$ and $(\mathcal{N}-1)N$ atoms of other components with strength $G$. Hence we expect that, on average, the intracomponent interaction strength will be suppressed by a factor of $\mathcal{N}-1$ (assuming $N \gg 1$) compared to the intercomponent one, yielding an average interaction strength $\propto (G + \frac{g}{\mathcal{N}-1})$. We anticipate the stability of this effective single-component mixture to be ensured as long as its average interaction strength is repulsive (i.e. $G \geq -\frac{g}{\mathcal{N}-1}$), since the effective single-component picture does not impose any limitations at the repulsive side. Indeed this pattern appears to hold for $\mathcal{N} = 2$, (see Sec. 3.1) $\mathcal{N} = 3$, (see Sec. 4.1) and $\mathcal{N} = 4$ (see Sec. 5.1), where the lower stability bound corresponds to $G \geq -g$, $G \geq -g/2$, and $G \geq -g/3$ respectively. On the other hand, the only upper bound imposed to the intercomponent interaction strength $G$, stems from the stability of its sub-systems which requires $G < g$ regardless of the number of components. This means that at the repulsive side all components will either be stable and fully overlap or all components will be mutually unstable and phase separate from all other components. A hypothetical partially phase separated configuration, with $\mathcal{M}$ overlapping components in one cluster and $\mathcal{N} - \mathcal{M}$ in the other, would feature average intra-cluster

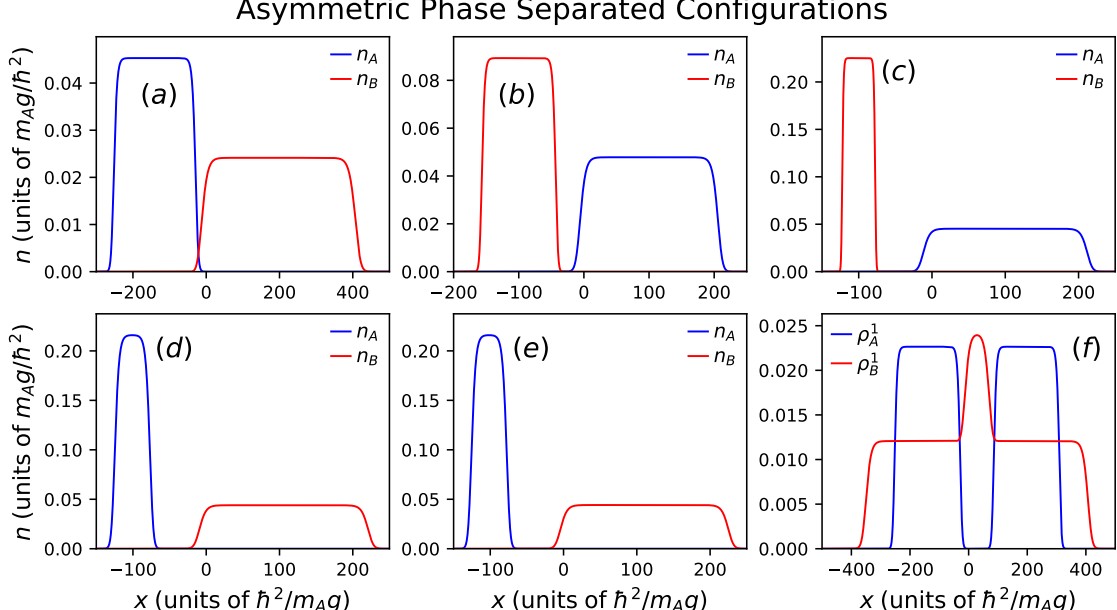

Figure 7: Parity asymmetric density configurations corresponding to (a) a homonuclear two-component mixture with interaction strengths $g_B = 0.5g_A$ and $g_{AB} = 0.4g_A$ and (b), (c) a heteronuclear mixture characterized by $g_B = g_A = g$, $g_{AB} = 0.4g$ and $m_B = 2m_A$ and $m_B = 5m_A$ respectively. (d), (e) The same as before but for a homonuclear three component mixture with two identical components featuring $g_A = g_B = g_C = g$, $g_{AB} = -0.5g$ as well as $G_C = 0.1g$ and $G_C = 0.4g$ respectively. (f) Reduced one-body density of the many-body wavefunction occupying the states corresponding to the classical-field solutions given in (a). In all cases each component contains $N_\sigma = 10$ atoms.

interaction strengths $\propto G + \frac{g}{\mathcal{M}-1}$ and $\propto G + \frac{g}{\mathcal{N}-\mathcal{M}-1}$ respectively, both of which are larger than the inter-cluster coupling strength $G$ (considering only repulsive interactions). As such, there is no reason for the clusters to phase separate, which explains why such configurations do not seem to appear.

## B  Asymmetric phase-separated configurations

The eGPEs derived in the main text support also ground state solutions with spontaneously broken parity symmetry in the case of mass-, or interaction-imbalanced mixtures featuring inter-component repulsion. Namely, these solutions are present in the cases where the phase separated configurations discussed in the main text appear as illustrated in Fig. 2 (c), Fig. 3(d), (f), and Fig. 5(e), (f). In Fig. 7(a)-(e) we present the corresponding parity asymmetric solutions for all cases mentioned above. In all cases, these solutions feature slightly lower energies (e.g. the relative energy difference is $\approx 0.3\%$ [$\approx 0.25\%$] for the homonuclear [heteronuclear with $m_B = 5m_A$] two-component mixture presented in Fig. 7(a) [Fig. 7(c)]) than the parity symmetric configurations shown in the main text. However, as mentioned in the main text, we consider such asymmetric phase-separated configurations to be associated with the classical-field approximation since they are not in principle permitted under a many-body wavefunction Ansatz. As such, within the main text we present and discuss the respective parity symmetric solutions as the ground state of the system in-spite of the slightly larger energy owing to the

larger kinetic energy. We have confirmed that both types of solutions (i.e., symmetric and asymmetric) are robust when exposed to small amplitude perturbations, e.g., emulated by a random noise distribution across the spatial profile of the solution, and they afterwards let to evolve at least for total evolution times $T = 10^4 \, \hbar^3/(m_A g_A^2)$ that we have checked. It is also important to remark that the existence of these configurations should not be under-appreciated. They can naturally emerge in corresponding experiments e.g. even in the presence of relatively small magnetic field gradients or other symmetry breaking contributions due to un-avoidable imperfection sources. Moreover, these domain-wall like distributions are interesting on their own right e.g. for dynamical applications in order to create topological excitations or study hydrodynamic instabilities.

To clarify our argument about the many-body wavefunction structure, consider as an example the homonuclear two-component mixture. Let $\Phi_A(x)$, $\Phi_B(x)$, be the single particle wavefunctions corresponding to the densities depicted in Fig. 7(a). We note in passing that all the configurations shown in Fig. 7(a)-(e) are, of course, doubly degenerate and their parity transformed counterpart is also a solution with the same energy, due to the parity symmetry of the ensuing Hamiltonian. Hence, neglecting any higher order correlations, the simplest form of the many-body wavefuntion for our homonuclear mixture, consistent with the results from the eGPEs would be:[6]

$$|\Psi\rangle = \sqrt{\frac{N_A N_B}{2}} \Big( |\Phi_A(x)\rangle |\Phi_B(x)\rangle + |\Phi_A(-x)\rangle |\Phi_B(-x)\rangle \Big). \tag{B.1}$$

Therefore, the corresponding diagonal of the reduced single-particle density matrix (namely, $\rho_\sigma^1(x) = Tr_\sigma \langle \Psi | \Psi \rangle$) of the underlying many-body system is not the one presented in Fig. 7(a), but it has the significantly more involved spatial profile illustrated in Fig. 7(f). Obviously, this structure can be associated with a significantly higher energy (at the level of the classical-field approximation) than either the parity symmetric solution shown in Fig. 2 (c), or the parity asymmetric solution depicted in Fig. 7(a). Finally, let us mention explicitly that the parity symmetric solutions discussed throughout the text can be directly associated with the reduced single-particle density of a many-body wavefunction (at the same level of approximation) since they are not degenerate. Namely, if $\Phi_A^S(x)$, $\Phi_B^S(x)$ are the parity symmetric single particle wavefunctions corresponding to the densities illustrated in Fig. 2(c), then $|\Psi\rangle = \sqrt{N_A N_B} |\Phi_A^S(x)\rangle |\Phi_B^S(x)\rangle$ is a valid (parity symmetric) many-body wavefunction, with the diagonal of its reduced single-particle density matrix being exactly the structure shown in Fig. 2(c). Thus, the densities of the parity symmetric solutions of the eGPEs can be interpreted as the reduced single-particle densities of the ground-state many-body wavefunction of the atomic gas and their energy can be interpreted as the energy of the many-body configuration (within the accuracy of the perturbative calculation). In contrast, parity asymmetric configurations do not permit this interpretation even if they correspond to a lower energy within the classical-field approximation, although upon proper symmetrization they could correspond to a valid (in general excited) many-body configuration (as the one shown in Fig. 7(f)). Of course as mentioned already, the situation changes in the presence of external parity breaking contributions which would lift the degeneracy of the parity asymmetric configurations and void the need for the symmetrization, thus making the corresponding parity asymmetric configurations valid candidates for the ground-state of the atomic gas.

---

[6]In general, the mirror symmetry is around the center-of-mass of the system ($c = \frac{1}{2}(\frac{1}{N_A} \int n_A x dx + \frac{1}{N_B} \int n_B x dx) \approx 28.5$ in our case) which is arbitrary owing to the underlying translation symmetry.

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
