# Peer review of "Multicomponent one-dimensional quantum droplets across the mean-field stability regime"

_SciPost Physics, doi:SciPost Phys. 19, 133 (2025)_

## Round 2 · Referee Report · Anonymous (Referee 1) · 2025-7-1

Strengths

1- Comprehensive analysis of the effect of beyond mean-field (BMF) terms in the equilibrium 1D configurations of binary mixtures of quantum gases

2- Comparison with previous analysis, whose validity was justified only for specific values of the interspecies interaction strength but used also elsewhere

3- Analysis of little explored mixtures of three and four components

Weaknesses

1 - In the three component mixture, the MF stability region is asymmetric: -g/2<G<g. No insight is given about this asymmetry.

Report

The manuscript reports an in-depth analysis of BMF quantum mixtures in 1D. The main merit is the derivation of LHY term valid for a large range of interspecies interactions, not only close to the critical points of MF collapse and immiscibility. The extension to three and four components is valuable, although inevitably limited to a few specific configurations, due to the large number of parameters (masses, interaction strength). In terms of usability for experiments, a limitation could be traced in the restriction to the homogenous case. However, in my opinion this is a necessary and justified choice to keep the number of external parameters at a manageable level. I see that this work has a clear potential for multi-pronged follow-up work. I recommend publication.

Requested changes

1- Page 2, in the sentence The emergence of the self-bound liquid-type configurations known as quantum droplets in contact interacting three-dimensional (3D) bosonic mixtures'', the reference [24] should be complemented with [44], as the latter was actually published earlier. 2- Page 4, sec. 2.1: it is stated that 1D configurations with periodic boundary conditions, i.e. a 1D ring,\dots can be experimentally reached with digital micromirror devices''. This is not correct, DMD can make rings, but these are hardly 1D, as the transverse confinement is insufficient. Also, Ref. [69] describing a disk-shaped potential seems quite not appropriate here. 3- Page 4, sec 2.1: dimensionally, $\hbar^3/(m_A g_A)$ cannot be a unit of time; 4- Page 5, below eq. 3: LaTeX, \verb|$diag$| should be typed as text, I think.

Recommendation

Publish (easily meets expectations and criteria for this Journal; among top 50%)

---

## Round 3 · Referee Report · Anonymous (Referee 2) · 2025-7-22

Report

This manuscript is comprehensive and clearly written, and I think will serve as a textbook for people working in the area of one-dimensional ultracold droplet physics, both theoretical and experimental. I find that the authors have adequately address the comments of the original referee's (very positive) review. I see no reason not to publish this paper as is.

Recommendation

Publish (meets expectations and criteria for this Journal)

---

## Round 3 · Referee Report · Anonymous (Referee 3) · 2025-8-10

Strengths

1- Thorough and lucid, this paper can be read as a guide to specific calculations or as a reference as is for future work 2- Detailed examples are presented, and the most important scenarios explored

Weaknesses

1- While the content lends itself well as a reference for future work, readers would likely benefit from a succinct summary of results, since it is easy for a reader to get lost in the sheer amount of parameters involved.

2- Only minor points, detailed in Requested Changes section.

Report

The authors consider the analytic form of the beyond mean-field Lee-Huang-Yang (LHY) correction in 1D condensates for two-, three-, and four-component mixtures under a variety of situations. The authors also consider numerical and analytical evaluations of the resulting ground state including these terms in the context of mixtures, leading to droplet formation prior to the mean-field instability being reached. As the number of free parameters dramatically increases as more components are added, the authors largely focus on highly-symmetric scenarios.

Overall, I view this work as a valuable resource for future analytic and numeric studies of mixtures in 1D. Since the LHY correction affects how these systems will behave near instability, it will be a valuable resource in understanding how mixtures behave in 1D, as well as better understanding quantum fluctuations in low dimensions. I recommend publication as it fits within Scipost's expectation of opening a new pathway in an existing research direction, with clear potential for multi-pronged follow-up work.

Requested changes

1- On Pg. 1, the authors write: “In 1D that we focus herein, the LHY term is attractive [53] instead of being repulsive as in 3D. As a result, the 1D droplet parametric regions i) do not feature collapse […] within the respective MF stability regime.” As written, this is confusing, as it seems to imply that an attractive LHY correction leads to stabilization, rather than being stabilized by the 1D geometry (e.g. KE term ~1/L^2) vs. the low powers of |psi| in the corrections. Upon reading the rest of the manuscript, it’s apparent the authors do not intend this meaning, but rather the “early” onset of droplets when compared to the non-LHY case. I would suggest changing these sentences.

2- The ground-state phases including three symmetric droplets (e.g. Fig 2 (c) ;Fig 3 (d), (f); Fig. 5 (e), (f)) give me a slight pause w.r.t. whether they are the true ground states. If the system separates into flat-topped droplets, why would one component split into two separated droplets, heightening its kinetic energy, rather than a single long droplet of the same average density? On P.g 12-13 the authors argue that the heavier component splits (rather than the lighter component) to reduce kinetic energy, but it’s not clear why either should split at all. In the case of even weak confinement [even weaker than mentioned in the footnote 4], then I would agree, the authors remind us we are in free space, I can’t see why we have this symmetric splitting.

Recommendation

Publish (meets expectations and criteria for this Journal)

---

## Round 3 · Referee Report · Anonymous (Referee 4) · 2025-8-13

Disclosure of Generative AI use

The referee discloses that the following generative AI tools have been used in the preparation of this report:

I used the AI Assist tool of overleaf to help english readibility.

Strengths

1) The authors provide an analytical derivation of the Lee-Huang-Yang (LHY) energy from two- to four-component one-dimensional bosonic mixtures, valid across the full Mean-Field stability regime. 2) Solving numerically the resulting extended Gross-Pitaevskii equations, they show quantitative deviations from the original MF-boundary LHY approximation.
3) The three- and four-component derivations pave the way for exploring exotic droplet phases.

Weaknesses

1) When considering Hetero-nuclear two-component mixtures, the authors compute numerically the LHY energy density (Eq. 11) to solve the eGPEs. They then compare to the exact equal-mass eGPEs (Eq. 14) with mass-imbalanced inserted by hand. The results perfectly agree (Fig. 3), but the authors do not comment on the reason behind this agreement.

Report

In their manuscript, the authors analytically derived the Lee-Huang-Yang energy for unconfined one-dimensional bosonic mixtures. They work goes beyond the case of two-component mixture by including the three- and four-component cases, and crucially is valid over the entire mean-field stability regime.

The authors numerically solve the resulting eGPEs to explore the ground-state properties of different interaction regimes and mixtures. They show substantial deviations from the original MF-boundary approximation. As the number of components increases, the parameter space grows drastically and the authors present a few specific scenarios. An interesting result is that the effect of quantum fluctuations is enhanced when the number of components increases.

The manuscript is well written and pedagogical.

Requested changes

1) p.8 "calculations [40–43]or experimental" -> missing space 2) Fig. 1. In the caption it could be specified that g_{AB} is given in unit of g g_{12} should be replaced by g_{AB} 3) p.10 g_{12} is used instead of g_{AB} many times

Recommendation

Publish (easily meets expectations and criteria for this Journal; among top 50%)

---

## Round 3 · Author Response

Dear Editor-in-charge of SciPost Physics,

Thank you very much for your communication on the above manuscript submitted to SciPost Physics. We are pleased with the positive receptions of our work by the referee. For instance, the referee remarks: "I see that this work has a clear potential for multi-pronged follow-up work. I recommend publication." Also, the recommendation of the referee is: "Publish (easily meets expectations and criteria for this Journal; among top 50%)".

Yet, the reviewer offers some helpful remarks towards the improvement of the manuscript. In the presently resubmitted revised version of our work, we have taken all of these comments into consideration and performed the corresponding changes. In the following, you can find our detailed reply to all questions and suggestions raised by the referee. A list of changes is appended after the point by point reply to the comments of the referee.

Given the positive outlook of the review, we hope that you will find the revised manuscript suitable for publication in SciPost Physics. We will, thus, be looking forward to your final editorial decision on the manuscript.

On behalf of all the authors, Ilias Englezos

  • The referee writes: "The manuscript reports an in-depth analysis of BMF quantum mixtures in 1D. The main merit is the derivation of LHY term valid for a large range of interspecies interactions, not only close to the critical points of MF collapse and immiscibility. The extension to three and four components is valuable, although inevitably limited to a few specific configurations, due to the large number of parameters (masses, interaction strength). In terms of usability for experiments, a limitation could be traced in the restriction to the homogenous case. However, in my opinion this is a necessary and justified choice to keep the number of external parameters at a manageable level. I see that this work has a clear potential for multi-pronged follow-up work. I recommend publication."

Our response is: We thank the referee for their positive evaluation of our work and useful comments which helped us to improve and further clarify our presentation. Below, we address in detail all the issues raised. Corresponding modifications have been performed in the revised manuscript and for convenience a list of changes is also appended at the end of this letter.

  • The referee writes: "1- Page 2, in the sentence The emergence of the self-bound liquid-type configurations known as quantum droplets in contact interacting three-dimensional (3D) bosonic mixtures'', the reference [24] should be complemented with [44], as the latter was actually published earlier."

Our response is: We agree with the referee. In the revised version of the manuscript, we have also placed this reference at the suggested location (see also the list of changes).

  • The referee writes: "2- Page 4, sec. 2.1: it is stated that 1D configurations with periodic boundary conditions, i.e. a 1D ring,\dots can be experimentally reached with digital micromirror devices''. This is not correct, DMD can make rings, but these are hardly 1D, as the transverse confinement is insufficient. Also, Ref. [69] describing a disk-shaped potential seems quite not appropriate here."

Our response is: We agree with the referee that these DMD setups do not necessarily reach the 1D geometry considered in our work. In the revised manuscript, we have removed this sentence and added a comment about the persistence of our findings with box potentials (see also the list of changes).

  • The referee writes: "3- Page 4, sec 2.1: dimensionally, $\hbar^3/(m_Ag_A)$ cannot be a unit of time."

Our response is: We thank the referee for pointing out this typo. We have corrected it to $\hbar^3/(m_A g_A^2)$, which is indeed a unit of time (see also the list the changes).

  • The referee writes: " 4- Page 5, below eq. 3: LaTeX, \verb|diag| should be typed as text, I think.

Our response is: We agree with the referee and we have changed ``diag" to text format in the revised manuscript (see the list of changes).

  • Finally the referee points out the weakness: "In the three component mixture, the MF stability region is asymmetric: $-g/2\leq G\leq g$. No insight is given about this asymmetry."

Our response is: We thank the referee for their suggestion. We have included a new discussion in Appendix A, to provide an intuitive explanation for this asymmetry. An additional comment has been incorporated in the main text, see also the list of changes.

---

## Round 3 · List of Changes

List of changes in the revised manuscript:

1) In the second paragraph of page 2, Section 1, the reference Science 359(6373), 301 (2018) has been placed earlier. 2) In the first paragraph of page 4, Section 2.1, a sentence regarding the possible DMD experimental implementation has been removed and replaced with a comment on the persistence of our results with a box potential. 3) In the second paragraph of page 4, Section 2.1, a typo in the time units used throughout the text has been fixed. 4) In the first paragraph of page 5, Section 2.2, the term "diag" has been changed to text format as suggested. 5) Second paragraph of page 16, Section 4.1, a comment pointing to Appendix A has been added. 6) In the last paragraph of pages 23-24, Appendix A, a short discussion about the asymmetry of the mean-field stability regime in higher-component mixtures has been included. 7) All references have been updated whenever necessary. 8) On page 9, we fixed a typo in the legend of Figure 1. 9) On page 10, a typo has been corrected in the legend of Figure 2. 10) On page 12, a typo has been amended in the legend of Figure 3. 11) On page 18, a typo has been updated in the title of Figure 5.

---

## Round 4 · Author Response

Dear Editor-in-charge of SciPost Physics,

Thank you very much for your correspondence regarding the above manuscript submitted to SciPost Physics. We are pleased to receive the positive feedback from the referees regarding our work. The referees recommend in reports \#1 and \#2: "*Publish (meets expectations and criteria for this Journal)*", and in report \#3: "*Publish (easily meets expectations and criteria for this Journal; among top 50\%)*".

Yet, two of the reviewers offer some useful suggestions towards the improvement of the manuscript. In the revised version of our work, which we are currently resubmitting, we have addressed all of these comments and incorporated the appropriate modifications. Below, we include our detailed reply to all the requested changes proposed by the referees. A list of changes is appended after the point-by-point reply to the comments of the referees.

Considering the positive feedback from all the reviewers, we hope that you will find the revised manuscript suitable for publication in SciPost Physics in its current version. We thank you very much for your handling of our work and looking forward to your final editorial decision on the manuscript.

On behalf of all the authors,

Ilias Englezos

<brS>

**Report \#1 by Anonymous (Referee 2) –- SciPost Physics/2502.08392v3**

**The referee writes:**

**This manuscript is comprehensive and clearly written, and I think will serve as a textbook for people working in the area of one-dimensional ultracold droplet physics, both theoretical and experimental. I find that the authors have adequately address the comments of the original referee's (very positive) review. I see no reason not to publish this paper as is.**

**Our response is:**

We sincerely thank the referee for their positive evaluation of our work and time commitment to review our manuscript.

**Report \#2 by Anonymous (Referee 3) –- SciPost Physics/2502.08392v3**

**The referee writes:**

**The authors consider the analytic form of the beyond mean-field Lee-Huang-Yang (LHY) correction in 1D condensates for two-, three-, and four-component mixtures under a variety of situations. The authors also consider numerical and analytical evaluations of the resulting ground state including these terms in the context of mixtures, leading to droplet formation prior to the mean-field instability being reached. As the number of free parameters dramatically increases as more components are added, the authors largely focus on highly-symmetric scenarios.**

**Overall, I view this work as a valuable resource for future analytic and numeric studies of mixtures in 1D. Since the LHY correction affects how these systems will behave near instability, it will be a valuable resource in understanding how mixtures behave in 1D, as well as better understanding quantum fluctuations in low dimensions. I recommend publication as it fits within Scipost's expectation of opening a new pathway in an existing research direction, with clear potential for multi-pronged follow-up work.**

**Our response is:**

We thank the referee for their positive evaluation of our work and helpful remarks towards the improvement of our presentation. Below, we address in detail all comments of the referee. Corresponding changes have been performed in the resubmitted manuscript and a list of changes is also appended at the end of this letter.

**The referee writes:**

**1- On Pg. 1, the authors write: “In 1D that we focus herein, the LHY term is attractive [53] instead of being repulsive as in 3D. As a result, the 1D droplet parametric regions i) do not feature collapse […] within the respective MF stability regime.” As written, this is confusing, as it seems to imply that an attractive LHY correction leads to stabilization, rather than being stabilized by the 1D geometry (e.g. KE term $\propto 1/L^2$) vs. the low powers of |psi| in the corrections. Upon reading the rest of the manuscript, it’s apparent the authors do not intend this meaning, but rather the “early” onset of droplets when compared to the non-LHY case. I would suggest changing these sentences.**

**Our response is:**

We agree with the referee. In the revised version of the manuscript, we have rephrased this statement to clarify our meaning, see also the list of changes.

**The referee writes:**

**2- The ground-state phases including three symmetric droplets (e.g. Fig 2 (c) ;Fig 3 (d), (f); Fig. 5 (e), (f)) give me a slight pause w.r.t. whether they are the true ground states. If the system separates into flat-topped droplets, why would one component split into two separated droplets, heightening its kinetic energy, rather than a single long droplet of the same average density? On Pages 12-13 the authors argue that the heavier component splits (rather than the lighter component) to reduce kinetic energy, but it’s not clear why either should split at all. In the case of even weak confinement [even weaker than mentioned in the footnote 4], then I would agree, the authors remind us we are in free space, I can’t see why we have this symmetric splitting.**

**Our response is:**

We are grateful to the referee for this insightful comment which incentivised us to expand our analysis of the phase separated configurations and to address some oversights in the revised manuscript. The additional discussions included in the revised manuscript about the existence of further phase-separated configurations diversified the intriguing droplet phases revealed by our work.

First, the referee is correct in noting that a single domain-wall like configuration with one component located on the left and the other on the right with respect to $x=0$ (and hence without splitting of either component) is supported by the eGPEs and has a slightly lower energy (typically of the order of $0.3 \%$) than the segregated one that we present in the main text. Since there is no explicit symmetry breaking term in the considered Hamiltonian, however, such a configuration violates the parity symmetry of the system. As such from the perspective of a many-body wavefunction which should respect the symmetry is not a-priori permitted, while it can exist as a solution to the classical field eGPE equations. For these reasons, in the main text we focused exclusively on parity symmetric solutions, which can be directly interpreted as the reduced one-body density of an one-dimensional bosonic gas (up to the first order perturbation theory of course). For completeness we have included an appendix in the revised manuscript, where we address such asymmetric phase-separated configurations for both homonuclear and heteronuclear mixtures as well as for three-component ones. In this Appendix, we also clarify why they should not be considered as the ground state of the underlying many-body system and elaborate on symmetry considerations. This fact certainly does not under-appreciate the impact of these configurations which should be detectable in corresponding experiments (e.g. in the presence of small field gradients or other symmetry breaking contributions due to imperfections) and have interesting applications on their own right. We have also added references to this appendix and emphasized that we focus on parity symmetric configurations wherever appropriate in the main text (see also the list of changes).

Regarding the heteronuclear system addressed on pages 12-13, we agree with the comment of the referee about the preferential splitting of the lighter component.
We have fixed this error and updated Fig.3 of the main text, while double-checking and further benchmarking our simulations. To further interpret our results we have also included an analytical estimation regarding the kinetic energy of these separated droplet configurations, which we find to scale as $\sim m_\sigma g^2_\sigma$. This scaling explains why in parity symmetric configurations the lighter or more weakly interacting component splits preferentially to reduce the kinetic energy of the system. We have included this observation in the appropriate discussions of the revised manuscript (see also the list of changes).

**Report \#3 by Anonymous (Referee 4) –- SciPost Physics/2502.08392v3**

**The referee writes:**

**In their manuscript, the authors analytically derived the Lee-Huang-Yang energy for unconfined one-dimensional bosonic mixtures. They work goes beyond the case of two-component mixture by including the three- and four-component cases, and crucially is valid over the entire mean-field stability regime.**

**The authors numerically solve the resulting eGPEs to explore the ground-state properties of different interaction regimes and mixtures. They show substantial deviations from the original MF-boundary approximation. As the number of components increases, the parameter space grows drastically and the authors present a few specific scenarios. An interesting result is that the effect of quantum fluctuations is enhanced when the number of components increases.**

**The manuscript is well written and pedagogical.**

**Our response is:**

We are thankful to the referee for their comments and positive assessment. In what follows, we provide a point-by-point reply to all issues raised along with a list of changes at the end of the reply letter.

**The referee writes:**

**1) p.8 ``calculations [40–43]or experimental" $->$ missing space.**

**Our response is:**

We thank the referee for pointing out this typo. We have corrected it in the revised text (see also the list of changes).

**The referee writes:**

**2) Fig. 1. In the caption it could be specified that g\_\{AB\} is given in unit of g g\_\{12\} should be replaced by g\_\{AB\}.**

**Our response is:**

We agree with the referee and accordingly we have changed the captions and the legend of Fig.~1 (see also the list of changes).

**The referee writes:**

**3) p.10 g\_\{12\} is used instead of g\_\{AB\} many times.**

**Our response is:**

We thank the referee for bringing this typo to our attention. We have corrected it throughout the revised text (see also the list of changes).

---

## Round 4 · List of Changes

List of changes in the revised manuscript:

  1. In the third paragraph of Section~1, on pages 1 and 2, a sentence regarding the impact of the 1D geometry has been rephrased for clarity.
  2. In Figure 1, on page 9, the captions have been amended to reflect more clearly the employed units.
  3. In the legend of Figure 1, on page 9, a typo has been corrected.
  4. In the first paragraph of page 10, Section~3.1, a reoccurring typo has been corrected.
  5. In the last paragraph of Section~3.1, on page 11, we have expanded the discussion with an estimation of the inter-component interaction energy in the absence of phase separation and an estimation of the kinetic energy of droplet configurations.
  6. In the last paragraph of Section~3.1, on page 11, we have included a comment regarding the parity symmetry of the configuration and a reference to Appendix B.
  7. Figure 3 on page 12 has been updated to fix an error in the labeling of the components.
  8. In the third paragraph of Section~3.2, on page 13, the discussion, along with the footnotes at the bottom of the page, have been updated to reflect the corrected figure. A reference to Appendix B has also been added.
  9. In the last paragraph of Section~3.2, on page 13, a typo has been corrected.
  10. Appendix B has been added on pages 24-25 to address the asymmetric phase-separated configurations supported by the eGPEs.

---

## Editorial Decision

published